

# Glacier inventories reveal an acceleration of Heard Island glacier loss over recent decades

**Levan G. Tielidze[1,2], Andrew N. Mackintosh[1,3], Weilin Yang[3]**

[1]Securing Antarctica's Environmental Future, School of Earth, Atmosphere and Environment, Monash University, Clayton, VIC 3800, Australia

[2]School of Natural Sciences and Medicine, Ilia State University, Tbilisi 0162, Georgia

[3]School of Earth, Atmosphere and Environment, Faculty of Science, Monash University, Clayton, VIC 3800, Australia

**Abstract**

Glacier inventories provide baseline data for understanding and evaluating past, current, and future changes in glacier extent in response to climate changes. We present a multi-year, manually mapped glacier inventory for sub-Antarctic Heard Island, a remote glacier-covered volcano in the southern Indian Ocean. Glacier outlines are presented for 1947, 1988, and 2019, derived from large-scale topographical maps (1:50,000), cloud-free medium-resolution SPOT, and high-resolution Pléiades satellite orthoimages. ASTER and Pléiades digital surface elevation models for 2000 and 2019 were also used to determine topographic parameters for individual glaciers. Heard Island glacier area reduced from 289.4±6.1 km$^2$ in 1947 to 260.3±6.3 km$^2$ in 1988, further decreasing to 225.7±4.2 km$^2$ in 2019. The rate of annual glacier area loss between the two observation periods (1947-1988 and 1988-2019) almost doubled from −0.25% yr$^{-1}$ to −0.43% yr$^{-1}$. Glaciers on the eastern slopes of Heard Island experienced much higher retreat rates than glaciers elsewhere on the island. The maximum retreat observed between 1947 and 2019 was ~5.8 km for the east-facing Stephenson Glacier, where collapse of the terminus led to the formation of a large lagoon during recent decades. Surface debris cover on Heard Island glaciers increased from 7.0±6% (18.1 km$^2$) in 1988 to 12.8±5.5% (29.0 km$^2$) in 2019. We also observed an upward shift (4.2 m yr$^{-1}$) in the maximum elevation of debris cover from 285±20 m a.s.l. (above sea level) to 605±20 m a.s.l., during this time. Direct climate observations from Heard Island are scarce, but reanalysis climate data show that the decline in glaciers is associated with a rising temperature of 0.7°C over the last seven decades. Many questions about the behaviour of Heard Island glaciers remain unanswered. Our inventory dataset will be freely available in the GLIMS glacier database to facilitate further analysis and modelling of Heard Island glaciers.

**Correspondence:** Levan G. Tielidze (tielidzelevan@gmail.com)

**Keywords:** Heard Island, glacier inventory, remote sensing, debris cover, glacier retreat.





## 1 Introduction

Glaciers are a key component of the climate system and serve as sensitive indicators of climate change (Hock et al., 2019). Over recent decades, global glaciers—excluding the Greenland and Antarctic ice sheets—have contributed approximately 27 ± 22 mm to the rise in global mean sea level from 1961 to 2016. This represents 25 to 30 percent of the total sea level rise during that period (Zemp et al., 2019). Glacier retreat also has a fundamental impact on global water supplies and mountain hazards (Hock et al., 2019) and glacier inventories are essential for assessing all these impacts (Haeberli and Hoelzle 1995). Inventories include digitized glacier outlines and their morphological features, such as area, slope, aspect, and elevation, mapped from satellite images and Digital Elevation Models (DEMs). They provide crucial data for estimating geodetic mass balance (Shean et al. 2020), determining glacier volume (Farinotti et al. 2019), measuring surface velocity (Dehecq et al. 2019), and testing glacier models (Radić et al., 2014; Eis et al., 2021).

Given the scarcity of landmasses and climate observations in the Southern Ocean region, glacier-covered sub-Antarctic islands provide a unique window into the impacts of past, present, and future climate changes (e.g. Deline et al., 2024). Heard Island is such an example but because of its harsh climate and remote location, its glaciers remain relatively poorly studied compared to other mid-latitude glaciers in the Southern Hemisphere. A new inventory from this region offers critical data on glacier behaviour, such as area and terminus changes, which are essential for understanding how these glaciers are responding to climate changes and other drivers. Given the limited field-based research opportunities on Heard Island, a new glacier inventory also allows us to analyse past glacier conditions, improving our understanding of processes and providing a baseline for predicting future changes in this sensitive region.

Heard Island is an UNESCO World Heritage site due to its outstanding physical and biological features which are being affected by significant on-going climatic changes. As one of the only sub-Antarctic islands mostly free of introduced species, its ecosystems are particularly at risk from the impact of glacier retreat (HIMI Management Plan, 2014). Glacier inventory work will help in designing effective conservation strategies and managing protected areas to ensure the preservation of the biodiversity they support (Pockley, 2001; HIMI Management Plan, 2014).

Since a complete inventory of Heard Island glaciers has not been published for several decades (e.g. Budd, 2000; Allison and Thost, 2000; Thost and Truffer, 2008), this study aims to create a new inventory using remotely sensed glacier parameters. We use our analysis to assess how various climatic, morphological and topographic factors influence glacier recession on Heard Island, providing preliminary answers to the question of how and why Heard Island glaciers are changing.



## 2 Study area

### 2.1 Physical characteristics

Heard Island is the largest member of the Territory of Heard Island and McDonald Islands located at 53°05'S latitude and 73°31'E longitude at the southern edge of Indian Ocean, almost midway between western Australia and South Africa. Antarctica is the nearest continent, located ~1500 km to the south, while the relatively large Kerguelen Islands are ~450 kilometers to the northwest. Heard Island is approximately 40 km long from northwest to southeast, and only 20 kilometers wide from northeast to southwest (Figure 1).

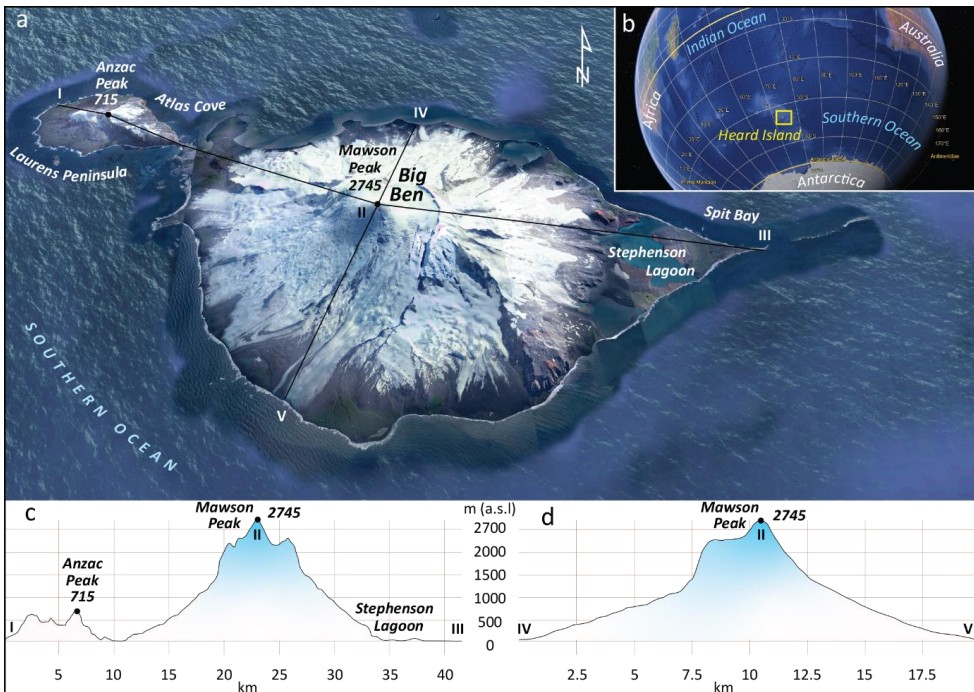

**Figure 1**. True colour aerial image of the Heard Island (© Google Earth). b - location of the Heard Island in the Southern Hemisphere (© Google Earth). c - Longitudinal and d - transverse profiles (cross sections) of Heard Island.

Heard Island, along with the McDonald Islands and Kerguelen Island, forms part of the submarine Kerguelen Plateau, a significant geological feature in the southern Indian Ocean. Heard Island's geological structure is characterized by its two volcanic cones — Big Ben and Laurens Peninsula — each with distinct elevations and volcanic rocks and landforms. Big Ben, with a diameter of ~20 km and reaching an elevation of 2,745 m asl at Mawson Peak, is the larger volcanic cone. The highest point of the Laurens Peninsula



is Anzac Peak (715 m asl). Over 80% of the Heard Island is ice-covered with glaciers
(Thost and Truffer, 2008) flowing from the caldera of Big Ben and the summit plateau of
Laurens Peninsula. Big Ben was most recently reported active in November 2020 (Fox et
al., 2021).
The climate of the Heard and McDonald Islands region is significantly shaped by its mid-
latitude position in the Southern Ocean, south of the Antarctic Polar Front, where
subantarctic and colder Antarctic Ocean waters converge. This location is affected by
the region's westerly winds, which are linked to the west-to-east progression of deep low-
pressure systems in the mid to high southern latitudes (Allison and Keage, 1986). Heard
Island's weather is characterized by small variations in both seasonal and daily
temperatures, with persistent cloud cover, frequent precipitation, and strong winds.
Snowfall occurs year-round. Based on various observations from the second half of the
20th century (Kiernan and McConnell, 2002; HIMI official website, last access: August
2024), average monthly temperatures at Atlas Cove range from 0.0 to 4.2°C. In summer,
daily temperatures typically vary between 3.7 and 5.2°C, while in winter, they range from
−0.8 to 0.3°C. The winds are predominantly westerly and strong, with monthly average
speeds at Atlas Cove varying between ~26 and 33.5 km/h. Gusts can exceed 180 km/h.
Heard Island experiences annual precipitation at sea level ranging from 1,300 to 1,900
mm, with rain or snow occurring on roughly three out of every four days. However,
meteorological records for the island are scarce and incomplete and long-term local
observations including information about local gradients and trends are absent.

**2.2  Previous studies**
Heard Island became widely known after 1853 when Captain Heard visited the site,
although it was first seen by Captain Peter Kemp in 1833 (Mawson, 1935). Two years after
Captain Heard's expedition, Darwin Rogers was the first to land on Heard Island in 1855
(Lambeth, 1951). Following this, sealers occupied the island from the mid to late 19[th]
Century, decimating the seal population. Later, various scientific expeditions visited
Heard Island and stayed only several days, mainly at Atlas Cove (e.g. Drygalski, 1908;
Aubert de la Rue, 1929; Mawson, 1932).
The largest scientific campaign began in 1947, when the Australian National Antarctic
Research Expedition established a base camp at Atlas Cove. This campaign continued
until 1955, and James Lambeth first carried out physical measurements of Heard Island
glaciers at this time (Lambeth, 1951). According to his notes the snow line of the Heard
Island glaciers was at about 305 m a.s.l. in Austral summer of 1947-1948. There was
minor difference in the snowline elevations between north- and south-facing glaciers.
Thickness of the Baudissin and Vahsel glaciers at the frontal area was between 35-40 m
in August 1948. Ice velocity measurement at an elevation of 90 m a.s.l. of Baudissin
Glacier tongue between 11 September and 8 December indicated an average ice flow



rate of approximately 440 m a⁻¹, while the ablation at an elevation of about 40 m a.s.l. was
~−0.35 m during this ~33-month period.
Short scientific visits to the island were also made in 1963, 1965 and 1969 mainly for
seasonal observation of the north-facing glacier front variations (Budd, 1964; 1970; Budd
and Stephenson, 1970). More detailed studies of Vahsel Glacier, including mass balance
measurements started later in 1971 (Allison 1980). Ablation at 200 m a.s.l. was estimated
at 3-4 metres water equivalent annually. A surface velocity profile across the glacier at
the same elevation showed annual movement of 200-280 m near the centre. Average
thickness of Vahsel Glacier (determined gravimetrically) below the equilibrium line, was
between 60-80 m near the centreline.
Studies of south and southeast facing glaciers starter later in 1990s. The retreat and
proglacial lake expansion at Stephenson Glacier was studied by Kiernan and McConnell
(2002), who showed that the glacier retreated at a mean rate of -18 ma⁻¹ from 1947 to
1987 before accelerating dramatically to -100 m a⁻¹ between 1987 and 2000. Further
glaciological and meteorological study of Brown Glacier on the eastern slope of Heard
Island was conducted by Thost and Truffer (2008) based on old topographical maps
(1947) and ground-based observations (2004). They concluded that Brown Glacier area
decreased by ~29% during the investigated period, while on average the surface lowered
by −0.50 m a⁻¹. This mass loss was consistent with interpolated summer (January-March)
temperatures in the area that indicated a +0.9 °C warming over the investigated period
(1947-2004). They also observed that the Brown Glacier terminus retreated by 63 m
between 2000 and 2003.
The GLIMS book (Kargel et al., 2014) identifies 29 glaciers on Heard Island with a total
area of 257 km² in 1988 (Cogley et al., 2014), while the Randolph Glacier Inventory (RGI6)
database identifies 31 glaciers with a total area of 254.4 km². Some glaciers in these
datasets have incorrect terminus positions, specifically the eastern-facing glaciers, and
nearly all glaciers are also characterized with inaccurate ice margins (due to
misidentification of ice divides). This issue persists in the RGI7 database (Maussion et al.,
164    2023).


**3  Data sources**
Conducting large-scale field-based glacier research on Heard Island is extremely
challenging due to topographic, logistical, financial, and safety obstacles (e.g. Allison
and Thost, 2000). Consequently, old topographical maps and remotely sensed images
from historical and current Earth observation platforms provide the most viable method
for tracking changes in glacier parameters (e.g. Tielidze, 2016; Freudiger et al., 2018;
Weber et al., 2020).



### 3.1 Old maps

The first aerial photographs of Heard Island glaciers were captured in 1947 using a hand-held 6" x 6" Fairchild F24 camera aboard an RAAF Walrus amphibious aircraft. These photos provided limited oblique coverage of the east and north coasts. However, a ground triangulation survey conducted in 1948, which focused on Laurens Peninsula and the north and east coasts, allowed for more accurate mapping of terrain features, including glacier fronts (Allison 1980). A large scale (1:50,000) topographical map of Heard Island (1964) shows glacier boundaries from 1947-1948 based on these early photographs and surveys (Division of National Mapping, 1964). Therefore, the glacier extent on Heard Island in 1947-1948 can be directly referenced from this topographical map, which reflects the original positions of the glaciers as depicted in the aerial images (Figure 2; Table 1).

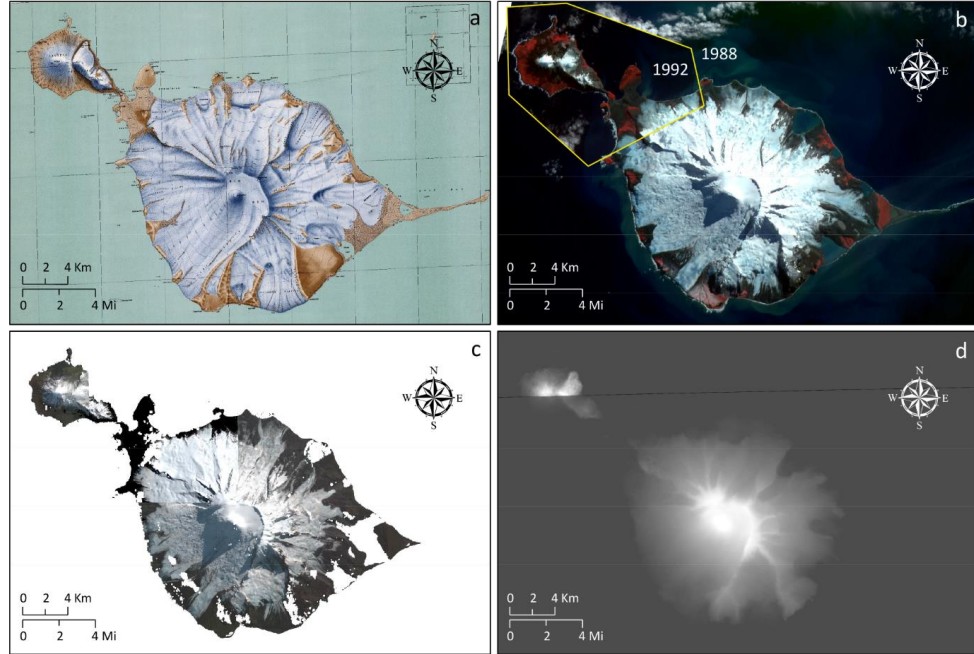

**Figure2**. Data sources used in this study for glacier mapping. a – Topographical map from 1964 compiled based on aerial imagery from 1947. b – SPOT images from 1988 and 1992. c – Pléiades images from 2019; d – ASTER DEM from 2000.

### 3.2 Satellite images

Although Landsat satellite products have been available since the 1980s, the frequent cloud cover around Heard Island has obscured the glaciers partially or completely, making it difficult to accurately delineate glacier boundaries. We therefore use medium-



resolution (10 and 20 m) cloud-free SPOT 1 images from 1988 and 1992 available from
the French Space Agency (CNES)'s Spot World Heritage Programme
(https://regards.cnes.fr/html/swh/Home-swh3.html, accessed August 2024). The 1988
image was used as the baseline database, and the 1992 image was used as a supplement
because the Laurens Peninsula and northern parts of the Heard Island glaciers were
partly covered by clouds in 1988 image.
A 2-metre resolution, georeferenced, cloud-free Pléiades ortho mosaic and 0.5-metre
resolution panchromatic bands from 2019 austral summer were used to delineate the
most recent glacier boundaries in our data set. These images were not available for the
northwest part of the Laurens Peninsula. To fill this gap, we used similar Pléiades product
acquired in April 2018, but the images from 2019 were used as a main dataset for the
island. Pléiades images were received from the Laboratory of Space Geophysical and
Oceanographic Studies (LEGOS) via the Pléiades Glacier Observatory (Berthier et al.,
209    2024).

High-resolution QuickBird images (2014-2024), combined with SRTM3 topography data
(Raup et al., 2014), were also used in Google Earth Pro software to enhance the 3D
visualization and identification of ice divides as well as to validate the glacier outlines
mapped from the satellite images.
For the annual measurements of Stephenson and Brown glacier termini, a 30 m
resolution Landsat 7 Enhanced Thematic Mapper Plus (ETM+) and 10 m resolution
Sentinel 2 images were used, as they only had cloud-free windows for these two glaciers.
Both Landsat and Sentinel products are from the Earth observation (EO) Browser
(https://apps.sentinel-hub.com/eo-browser/ accessed August 2024).

### 3.3 Digital elevation/surface models

Topographic parameters of glaciers such as aspect, slope, and elevation from 1947 and
1988 were obtained from the Advanced Spaceborne Thermal Emission and Reflection
Radiometer Global Digital Elevation Model (ASTER GDEM V2, March 1, 2000), available
through the EarthExplorer (http://earthexplorer.usgs.gov/, accessed May 2024), U.S.
Geological Survey (USGS), with a horizontal resolution of 20 meters (Tachikawa et al.,
2011). The ASTER GDEM is widely recognized and used in glaciological research globally
across different spatial scales (e.g. Bhambri et al. 2011; Stokes et al., 2013; Tielidze et
al., 2020). The same topographic characteristics for the glaciers from 2019 were
extracted from the high-resolution (2 m) Pléiades Digital Surface Models (DSM) 2018-
2019 available from the Laboratory of Space Geophysical and Oceanographic Studies
(LEGOS) via the Pléiades Glacier Observatory (Berthier et al., 2024).



**Table 1**. Topographic maps, satellite images and digital elevation/surface models used in
this study.

| Product | ID | Date | Resolution/Scale |
|---------|-----|------|------------------|
| **Map** | | | |
| Topographical map | G9182.H4 | 1964 (1947) | 1:50 000 |
| **Satellite image** | | | |
| SPOT 1 | 1 245-457 88-01-09 04:47:36 1 X | 09/01/1988 | 10 m |
| SPOT 1 | 2 245-458 92-03-24 05:00:31 1 X | 24/03/1992 | 20 m |
| Landsat 7 ETM+ | LE07_L2SR_135097_20010131_20200917_02_T1 | 31/01/2001 | 30 m |
| Landsat 7 ETM+ | LE07_L2SR_135097_20030105_20200916_02_T1 | 05/01/2003 | 30 m |
| Landsat 7 ETM+ | LE07_L2SR_135097_20040312_20200915_02_T2 | 12/03/2004 | 30 m |
| Landsat 7 ETM+ | LE07_L2SR_135097_20050126_20200914_02_T1 | 26/01/2005 | 30 m |
| Landsat 7 ETM+ | LE07_L2SR_135097_20080204_20200913_02_T1 | 04/02/2008 | 30 m |
| Landsat 7 ETM+ | LE07_L2SR_135097_20090206_20200912_02_T1 | 06/02/2009 | 30 m |
| Landsat 7 ETM+ | LE07_L2SR_135097_20100209_20200911_02_T1 | 09/02/2010 | 30 m |
| Sentinel 2 L2A | 42/F/YF/2017/4/16/0/ | 16/04/2017 | 10 m |
| Pléiades | 0444140_Heard_ANT_1B | 22/04/2018 | 0.5-2 m |
| Pléiades | 0451575_Heard_ANT_1B | 07/04/2019 | 0.5-2 m |
| Pléiades | 0452069_Heard_ANT_1B | 07/04/2019 | 0.5-2 m |
| Pléiades | 0448386_Heard_ANT_1A | 21/11/2019 | 0.5-2 m |
| **Digital elevation/surface model** | | | |
| ASTER | ASTGTMV003_S53E073 | 01/03/2000 | 20 m |
| ASTER | ASTGTMV003_S54E073 | 01/03/2000 | 20 m |
| Pléiades | 0444140_Heard_ANT_1B | 22/04/2018 | 2 m |
| Pléiades | 0451575_Heard_ANT_1B | 07/04/2019 | 2 m |
| Pléiades | 0452069_Heard_ANT_1B | 07/04/2019 | 2 m |
| Pléiades | 0448386_Heard_ANT_1A | 21/11/2019 | 2 m |


**4 Methods**
*4.1 Glacier mapping*
Despite some advantages of the automated mapping method of clean ice (e.g. Paul et
al., 2013), manual mapping of glaciers is more suitable for many mountain regions
around the world (e.g. Stokes et al., 2013; Nagai et al., 2016; Tielidze and Wheate, 2018;
Korneva et al., 2024). In this study, glacier boundaries were, therefore, delineated
manually. This was also mainly due to the i) unavailability of cloud-free satellite
channels/bands for different years for the entire study area, which limited us to use
different band ratio segmentation methods for automated mapping; ii) significant
amount of debris-cover, which can cause uncertainty during automatic mapping; and iii)
a relatively small study area, which was less time expensive than it would have been for
an entire mountain range.



### *4.2 Terminus measurement*

Accurately quantifying changes in glacier termini is essential for effective monitoring of glacier changes over various timescales, from days to centuries. Methods for this technique each offer advantages and limitations (Lea et al., 2014). In this study, we only measured two glacier (Stephenson and Brown) lengths based on the Global Land Ice Measurements from Space (GLIMS) guidelines (www.glims.com) and by following Purdie et al. (2014). The flow direction of the glacier was manually determined to be perpendicular to altitude contours. We assessed terminus changes by comparing glacier outlines from different dates along the ice front, oriented perpendicular to the flow. Elevation changes at the glacier fronts were also measured at the intersection points.

### *4.3 Accuracy and uncertainly assessment*

When using glacier outlines to assess changes, it's crucial to understand their accuracy. Assessing accuracy can be difficult because high-resolution reference data are often scarce. Additionally, any manual corrections made to the raw outlines (e.g. such as debris cover) may reflect the bias of the analyst rather than the effectiveness of the algorithm used (Paul et al., 2013). It is therefore essential to establish an uncertainty even after achieving high-accuracy mapping. Uncertainty often arises from the resolution of the satellite image, and from the contrast between the glacier and the surrounding terrain (Burgess and Sharp, 2004; DeBeer and Sharp, 2007). To estimate the statistical uncertainty for each glacier, we used the buffer method as in Granshaw and Fountain (2006) and Bolch et al. (2010). This approach provides a minimum and maximum area values, which we utilized to calculate the relative area difference. For the 1:50,000 scale topographic map, or glacier outlines from 1947, a 30 m buffer size proposed by Tielidze (2016) was used, yielding a total potential error of ±2.1%. A 20 m buffer was used for 1988 glacier outlines giving a total error of ±2.4%. The chosen buffer is based on a previous multiple digitizing experiment from similar resolution satellite images worldwide (Paul et al., 2013; Tielidze and Wheate, 2018), demonstrating that the variability in positioning typically falls within one pixel, or approximately ±10-30 meters, depending on the image resolution. Despite the high resolution of the Pléiades images, we choose a 10 m buffer size for glacier contours from 2019 estimating a total uncertainty of ±1.9%. This decision is primarily due to an extensive debris cover, which frequently complicates the mapping of glaciers.

A larger buffer should be applied to the debris-covered parts of glaciers, as ouline uncertainties are higher than for bare ice (e.g., Tielidze et al., 2020). In our case, the buffer size was set to 30 meters, resulting in a total potential error of ±6.0% for 1988 and ±5.5% for 2019 glaciers. We have not estimated the surface debris cover for the glaciers from 1947.



For uncertainties in length changes, we used the source image resolution as proposed by
Hall et al. (2003).

**5 Results**
**5.1 Glacier area changes in 1947-1988-2019**
A total of 29 glaciers, with an average size of 9.9 km², were mapped in the study area in
1947 (Figure 3). During the 41-year observation period from 1947-1988, the glaciers
shrank by 29.1±0.7 km² (10.1%) from 289.4±6.1 km² in 1947 to 260.3±6.3 km² in 1988,
showing a mean annual glacier area loss of −0.25% yr⁻¹. 30 glaciers were identified in
1988 with an average size of 8.7 km². Between 1988 and 2019, a considerable decrease
in glacier area of 34.6 km² (13.3%) to 225.7±4.2 km² occurred, almost doubling the rate
of annual glacier area loss to −0.43% yr⁻¹. The average glacier size also decreased to 6.4
km² during this period.

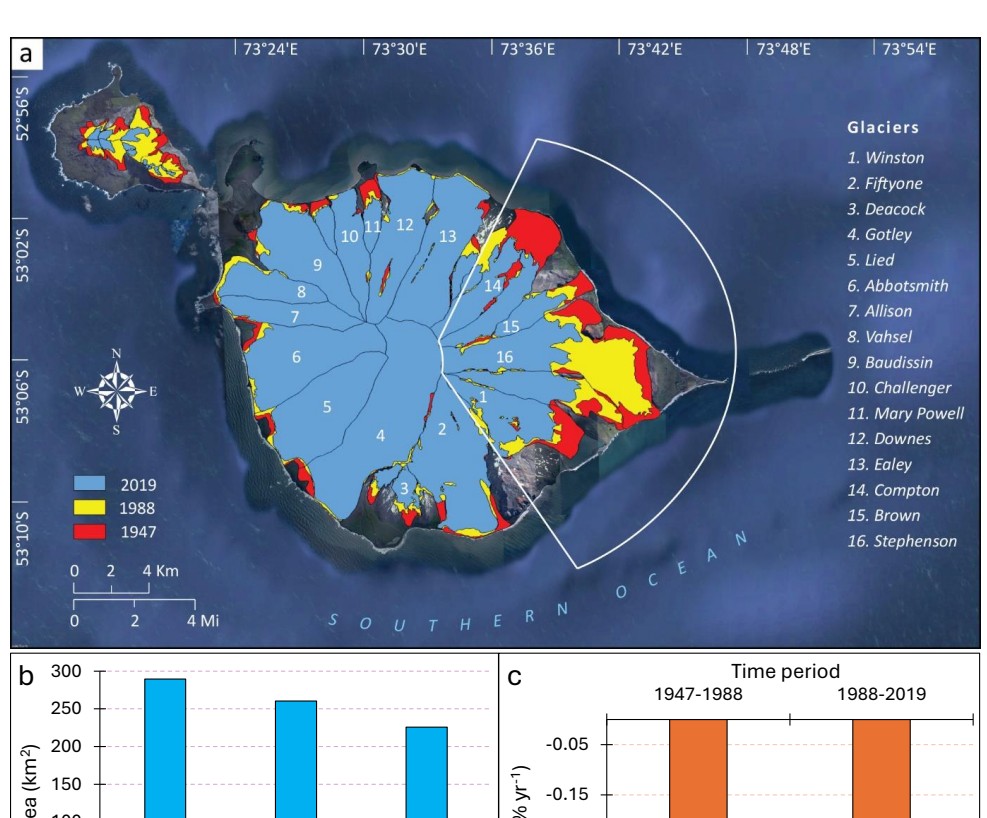


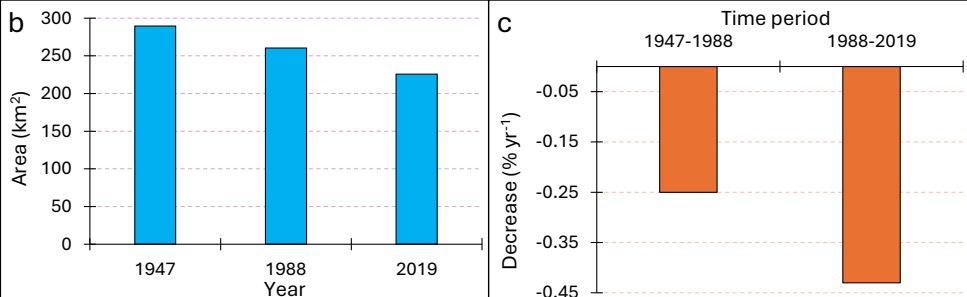






**Figure 3**. a - Heard Island glacier outlines in 1947, 1988, and 2019. The white frame highlights east-facing glaciers where the highest decrease rates occurred. b - Heard Island glacier areas in 1947, 1988, and 2019; c - Glacier area decrease rates for the Heard Island for the two observed periods (1947-1988 and 1988-2019).

The pattern of observed glacier wastage is strongly influenced by topographic and morphological parameters. The small and low elevation glaciers at Laurens Peninsula experienced largest change in area from 10.5 km$^2$ to 2.2 km$^2$ between 1947-2019. This is an area loss of 79±2.2% amounting to an annual decrease of −1.1% yr$^{-1}$.

We also observed upward shift of the minimum and mean elevation of all Heard Island glaciers during the study period and downward shift of the maximum elevation from 1988 (Figure 4).

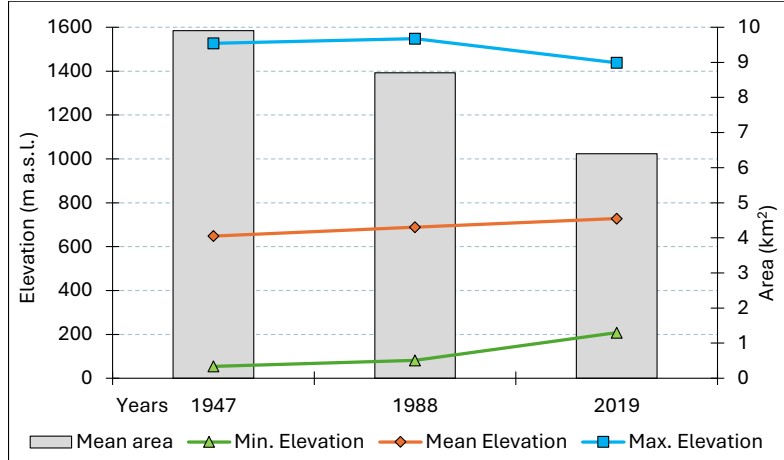

**Figure 4**. Changes in Heard Island glacier area and elevations between 1947-1988-2019. Mean area, minimum (Min.), mean, and maximum (Max.) elevations are shown.

Gotley Glacier was largest on Heard Island in 2019, with an area of 32.3 ± 0.5 km$^2$, followed by Fiftyone and Abbotsmith glaciers with areas of 21.8 ± 0.4 and 19.9 ± 0.3 km$^2$, respectively.

The inventory shows that Heard Island glaciers are predominantly oriented towards the northeast, while the largest glaciers are oriented southwest. Glaciers of west and southwest orientation have the highest mean elevations (Figure 5a-c).

Eastern facing glaciers experienced the highest area loss over the study period (depicted by the white outline in Figure 3a) from 85.1 ± 1.6 km$^2$ in 1947 to 66.5 ± 1.3 km$^2$ in 1988 and 46.3 ± 0.9 km$^2$ in 2019. This was a 22% or −0.53% yr$^{-1}$ decrease between 1947-1988 which




is more than double than the change for all Heard Island glaciers over this period. Glacier
change during the more recent investigated period was dramatic and unprecedented for
the eastern facing glaciers on Heard Island - they declined by 30% or −1.0% yr⁻¹ between
332    1988-2019.


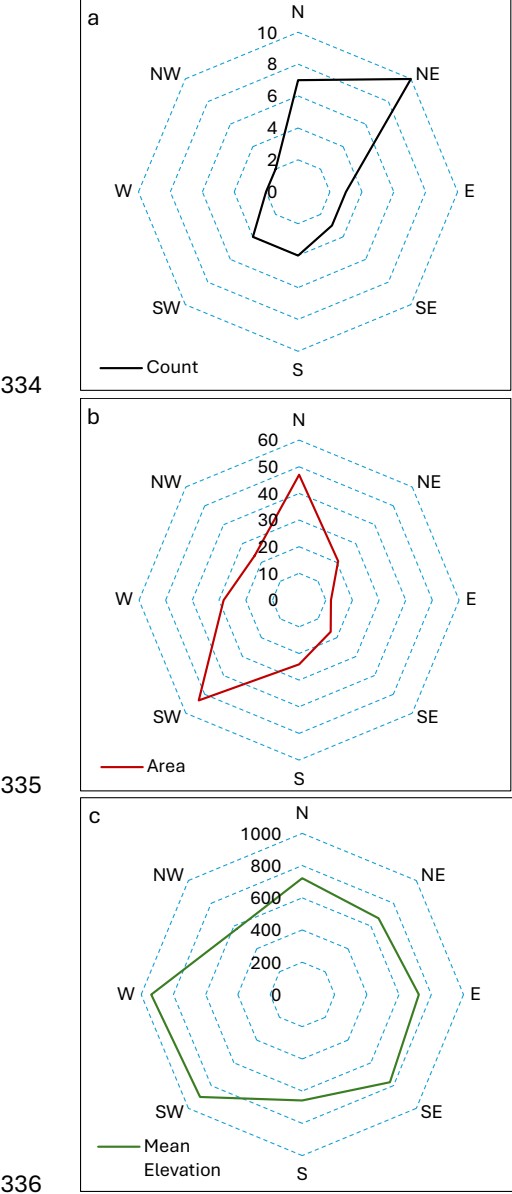




**Figure 5**. Distribution of Heard Island glacier aspects by (a) number, (b) area (km$^2$) and (c)
mean elevation (a.s.l.) in 2019.




## 5.2 Glacier length changes

Brown and Stephenson glaciers showed significant terminus retreat, as evidenced by
glacier shrinkage during the study period. Average total terminus retreat of 1483±20 m
was observed for Brown Glacier between 1947–2019 with an annual mean retreat rate of
−20.6 m yr⁻¹ (Figure 6a). The retreat rate of −25.7 m yr⁻¹ was much higher during the
second investigated period (1988-2019).

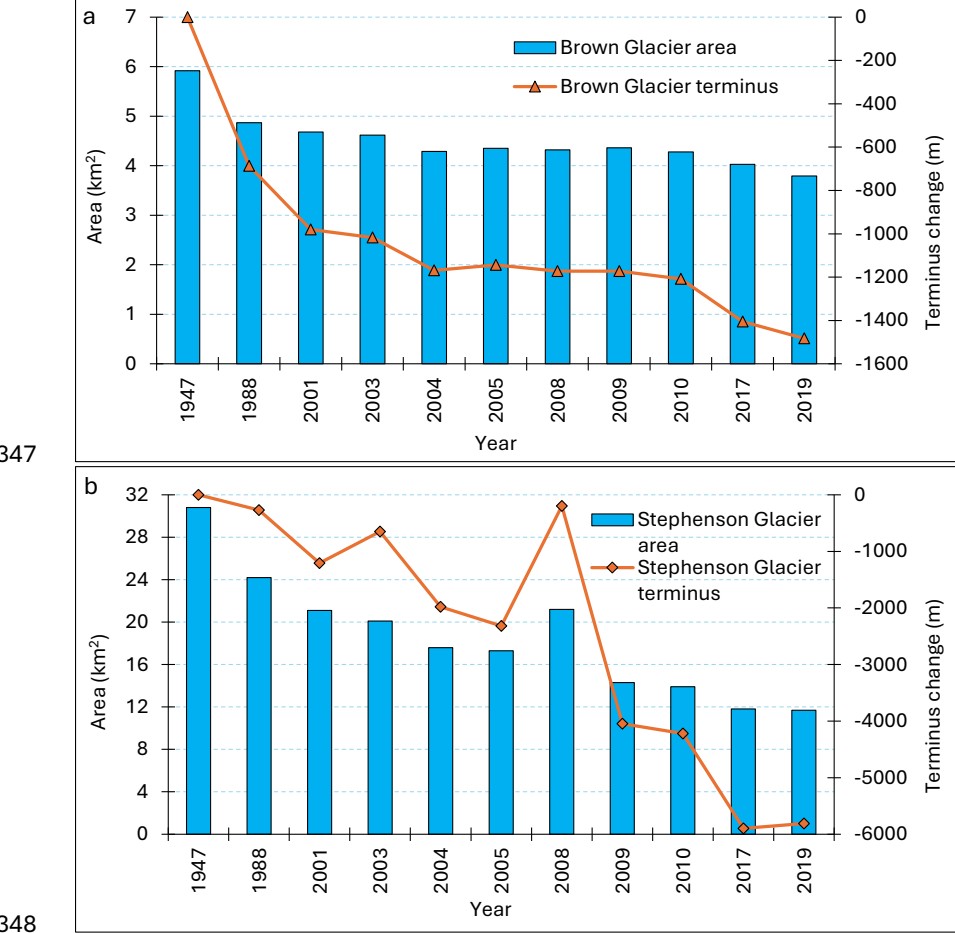



**Figure 6**. Cumulative terminus and area changes for Brown (a) and Stephenson (b)
glaciers between 1947 and 2019.

Stephenson Glacier retreated more than any other glacier on Heard Island during our
study period. The glacier terminus retreated by 5811±20 m between 1947–2019 yielding



an annual retreat rates of −80.7 m yr⁻¹ (Figure 6b). Despite experiencing small readvances
or relatively stable positions during some years, this didn't compensate for the massive
overall retreat during this 72-year period. Stevenson Glacier retreated by 5541±20 m
during the more recent retreat period (1988-2019), corresponding to an annual retreat
rate of −178.7 m yr⁻¹ which is seven times higher than it was observed for Brown Glacier
during the same time (Figure 6b). The largest readvance was in 2005-2008 when glacier
terminus extended by 2118±20 m but this was followed by a larger retreat of −3846±20 m
yr⁻¹ between 2008-2009 (Figure 7).

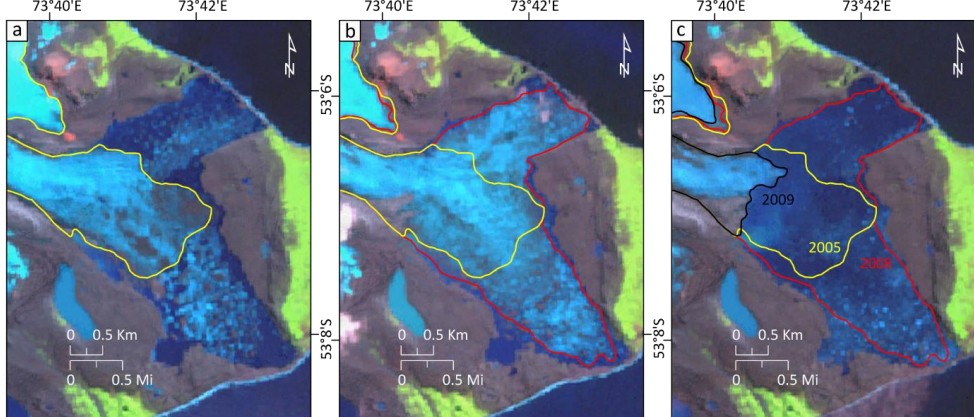


**Figure 7**. Stephenson Glacier advance and retreat in 2005–2008–2009. a – Landsat 7
ETM, 21/01/2005. b – Landsat 7 ETM, 04/02/2008. c – Landsat 7 ETM, 06/02/2009.

**5.3 Surface debris cover changes in 1988-2019**
We found that surface debris cover increased for all glaciers on Heard Island except
those facing northwest and for glaciers on the Laurens Peninsula. Overall debris cover
increased from 18.2±1.1 km² in 1988 to 29.0±1.6 km² in 2019. This increase in debris-
covered area occurred despite a large decrease in corresponding glacier area over this
period. The change equates to an increase in the proportion of debris-covered surface
area from 7.0±6.0 % in 1988 to 12.8±5.5 % in 2019 (Figure 8).
The mean upper limit (or maximum elevation) of the debris cover shifted from 285±20 to
605±20 m a.s.l. between 1988 and 2019 for all debris-covered glaciers on Heard Island.
The mean elevation of debris-covered parts also moved upwards from 165±20 to 410±20
m a.s.l. during this time. Southwest-facing Gotley Glacier had the highest mean upper
limit of surface debris cover with an elevation of 985±20 m a.s.l. Glaciers at Laurens
Peninsula had a small debris cover in 1988 but had become debris-free by 2019.



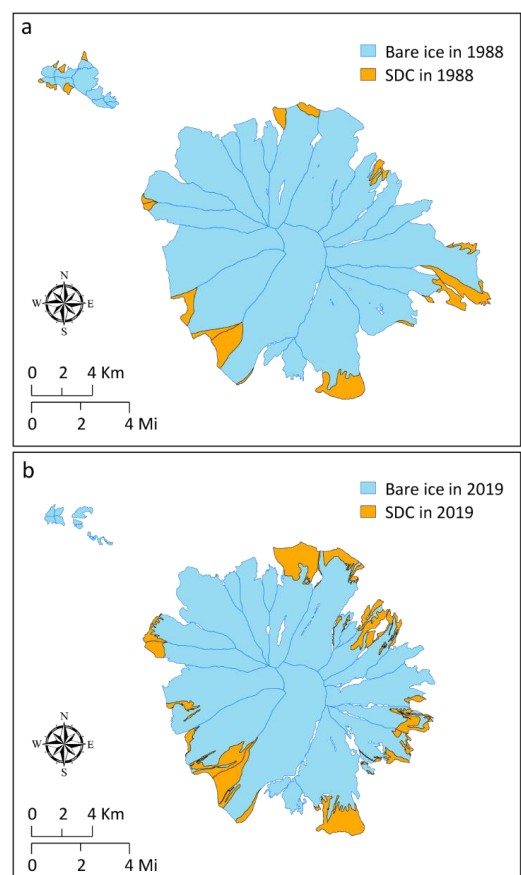


**Figure 8**. Decrease in bare ice and increase in surface debris cover (SDC) on Heard Island glaciers between 1988 (a) and 2019 (b).


## 6  Discussion

### 6.1 Glacier retreat and climate drivers

The relationship between climate and glacier changes in the Heard Island region must be considered cautiously because no local long-term weather station data are available. Instead, 2m surface air temperatures from ERA5 reanalysis (Soci et al., 2024) (0.5x0.5 deg) from a region encompassing Head Island (50-53° S, 70-73° E) were investigated between 1947 and 2019. Comparison between the first (1947–1959) and last (2009–2019) decades show summer temperature (NDJFM) increase by 0.7 °C which is consistent with previous record by Thost and Truffer (2008) who observed 0.8 °C increase of the summer temperatures between the 1948–1955 and 1997–2005 epochs. As shown in Figure 9a, a warming trend of the mean annual summer temperatures is also consistent with the increasing rate of glacier melting over the last 72 years. The period from 1980s onwards



is particularly notable for increasingly frequent and extreme positive temperature
anomalies relative to the long-term mean (Figure 9b).
Maritime glaciers such as those on Heard Island are particularly sensitive to air
temperature changes (e.g. Anderson and Mackintosh, 2006; Davies et al., 2014).
Warming causes the frequent precipitation to fall as rain rather than snow, decreasing
surface albedo over the glaciers and leading to increased absorption of short-wave
radiation and surface melt. Warmer temperatures also increase melt rates due to strong
turbulent exchange in these windy environments (e.g. Anderson et al., 2010; Anderson
and Mackintosh, 2012). For these reasons, the observed summer temperature increases
over the last 70 years are likely an important driver of ongoing glacier recession on Heard
Island. However, few precipitation measurements exist for Heard Island (Thost and
Truffer, 2008), and Favier et al. (2016) showed that retreat of the Cook Ice Cap on the
relatively nearby Kerguelen Islands is likely due to atmospheric drying since the 1960s
rather than atmospheric warming. Further observations and modelling are required to
provide more confidence in our interpretation that climate warming was largely
responsible for driving recent glacier retreat on Heard Island.
Our new glacier inventories indicate that Heard Island glaciers experienced overall
negative glacier mass balance between 1988 and 2019. This finding is supported by
satellite-based geodetic mass balance estimates for all Heard Island glaciers extracted
from the global study of Hugonnet et al. (2021) (Figure 10). However, a different mass
balance estimate derived from a mixed method that incorporates in situ observations
(Dussaillant et al., 2024) shows a positive geodetic mass balance trend between 1976
and 1998, in contrast to our observed glacier area reduction during the same period. This
mismatch (Figure 10) is likely due to the lack of direct mass balance observations to
constrain the estimates of Dussaillant et al. (2024) in this remote region.
The warming trend evident in ERA5 data in the Heard Island region (0.7°C) is relatively
small to the observed 1.1°C warming for the neighbouring Kerguelen Island between
1951-2020 (Nel et al., 2023) but it is in agreement with general Southern Ocean warming
trend since 1950s (Auger et al., 2021; Li et al., 2023). The primary driver of atmospheric
and ocean warming in this region is the human-induced shift towards the positive phase
of the Southern Annular Mode (Pohl et al., 2021), which has caused an intensification and
poleward shift of the Southern Hemisphere westerly winds (Perren et al., 2020). Further
accelerated decline in ice cover and warming air temperatures in the Heard Island region
mean that the glaciers may be rapidly approaching an irreversible tipping point in the sub-
Antarctic (Bakke et al. 2021), impacting landscape, geomorphic and ecosystem
processes.
Volcanic activity is another potential factor that could be contributing to the accelerated
glacier melting on Heard Island. Glacier retreat might also be a trigger for enhanced
volcanism at Heard Island due to decompression of the underlying magma chamber (e.g.
Barr et al., 2018). Eruptions could cause significant melting of the ice or deposition of



tephra, while increases in geothermal heat may lead to greater basal melting, thereby
influencing glacier movement (Allison and Keage, 1986; Fox et al., 2021). However, given
the lack of direct evidence in satellite images for unusual melt or tephra cover, the fact
that accelerated glacier melt occurred on all Heard Island glaciers and not just a subset,
and also on nearby Kerguelen Islands (Berthier et al., 2009; Deline et al., 2024), we
consider that any volcanic drivers of glacier retreat are less important than climatic ones.

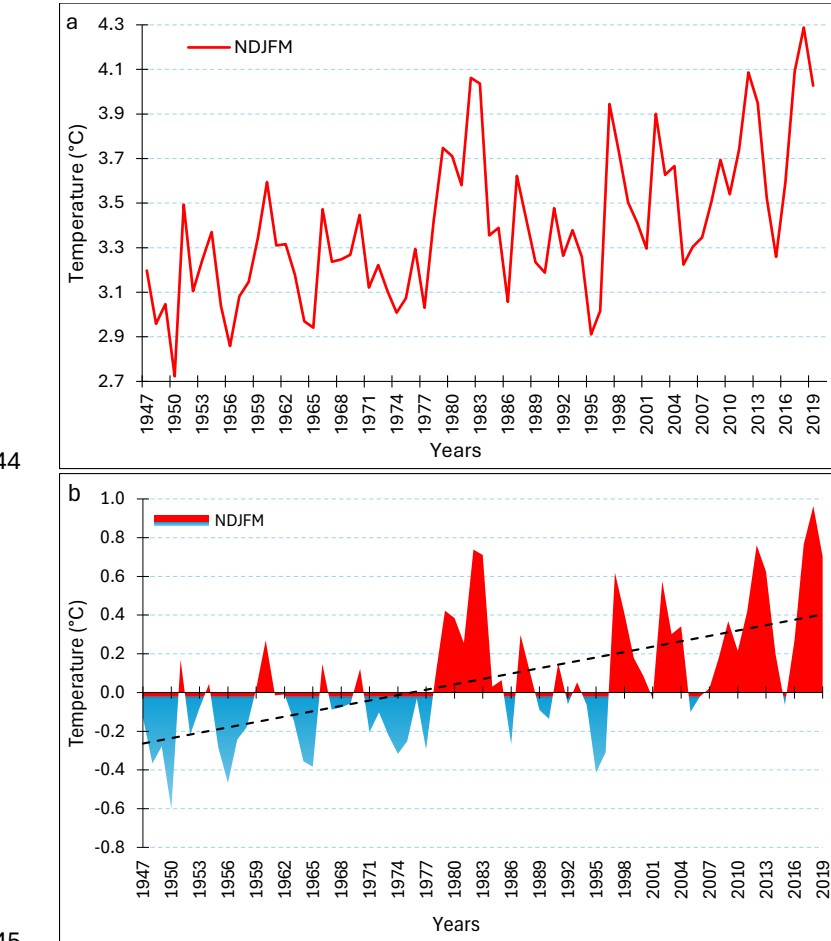



**Figure 9**. a - Mean annual warm season (NDJFM) temperatures (°C) for Heard Island
between 1947-2019. b - Time series of warm season (NDJFM) monthly air temperature
anomalies for the same period. The black dashed line on panel 'b' is the trend showing a
warming rate of 0.7°C over time. Both panels are based on the ERA5 2m surface air
temperature dataset.





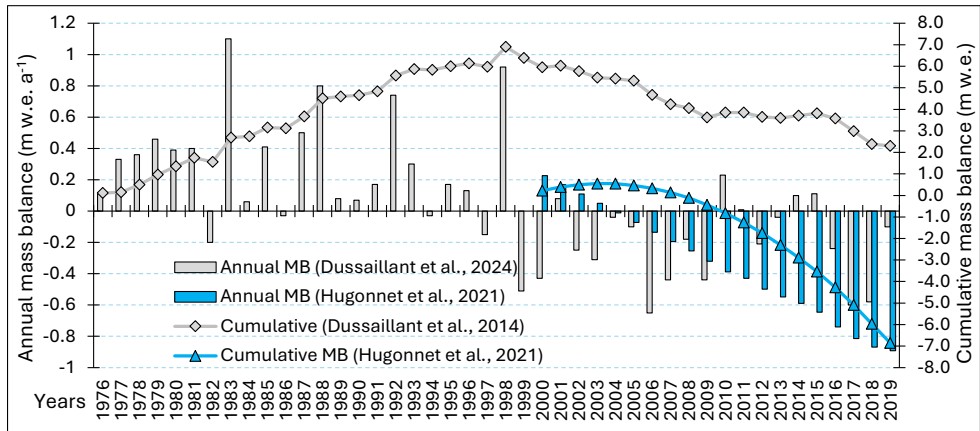

**Figure 10**. Mean annual geodetic mass balance (m w.e. a⁻¹) and cumulative mass
balance (m w.e.) for all Heard Island glaciers between 2000-2019 (Hugonnet et al., 2021)
and between 1976–2019 (Dussaillant et al., 2014).

## 6.2 The role of aspect and glacier lake formation on glacier retreat

We observed that higher rates of glacier retreat occurred on the eastern side of the island
(Figure 3). Although climate data are sparce and a process-based modelling approach is
required for further investigation, we expect this greater sensitivity of east-facing glaciers
is due to factors such as increased foehn winds (and correspondingly warmer air
temperatures) and reduced precipitation on the downwind side of the island. Our findings
agree with previous studies of Heard Island glaciers (e.g. Thost and Truffer, 2008) and
other studies in the sub-Antarctic region which show a similar asymmetry. For example,
Berthier et al. (2009) observed that the eastern part of Cook Ice Cap on Kerguelen shrank
2.5 times more (~28%) than the western part (~11%) between 1963 and 2003. Similar
observations have been made in South Georgia, where glacier decrease on the east coast
was higher than on the windy and wet southwest coast during the second half of the 20th
century (Gordon et al., 2008).
Comparison with other glacier retreat records from the southern mid to high latitudes
(Figure 11) shows that lake formation is a key driver of accelerated glacier retreat. Brown
Glacier on Heard Island initially experienced rapid retreat as a proglacial lake formed
between 1947 and 1988; this is like Chamonix Glacier on Kerguelen which initially
retreated in a lake but has been land terminating for several decades subsequently. The
retreat rate of these glaciers once they become land-terminating is relatively linear and
similar to the retreat rate of the land-terminating Fox and Franz Josef Glaciers in New
Zealand (Purdie et al., 2020; Mackintosh et al., 2017) (Figure 11).
In contrast, Stephenson Glacier on Heard Island is notable for its unprecedentedly high
terminus retreat, particularly between1988 and 2019 when the glacier retreated by





almost 5.5. km compared to 0.8 km at Brown Glacier (Figure 11). Lake formation at
Stephenson Glacier was much later than at Brown, and once a proglacial lake did form it
became extremely large. This is like the situation at Agassiz Glacier on Kerguelen which
retreated into its proglacial lake more recently than Chamonix Glacier (Deline et al.,
2024), and also Tasman Glacier in New Zealand which retreated more than 4.5 km in its
proglacial lake over recent decades (Mackintosh et al., 2017).

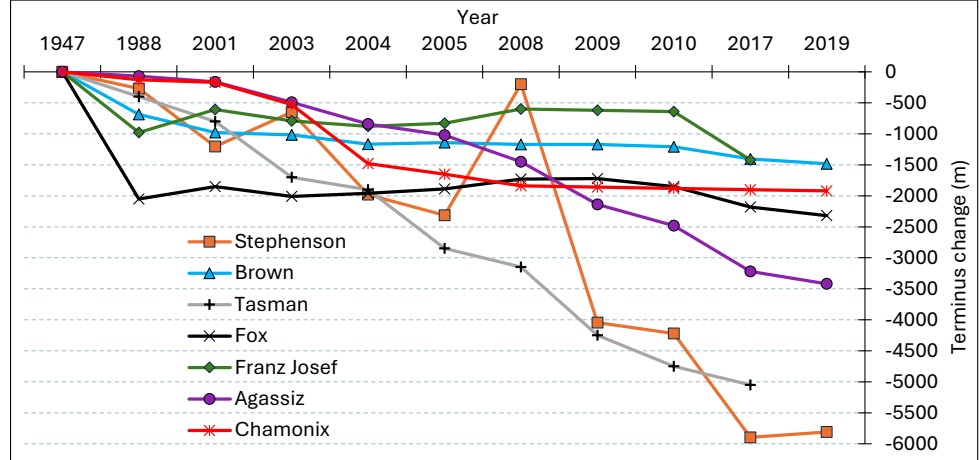


**Figure 11**. Cumulative terminus changes between 1947–2019 for Brown and Stephenson
glaciers (current study) compared to Agassiz and Chamonix glaciers on Kerguelen
(Deline et al., 2024) and Fox, Franz Josef and Tasman glaciers in New Zealand (Purdie et
al., 2020; Mackintosh et al., 2017).

The retreat of Stephenson Glacier warrants further discussion. In the 1950s, the glacier
had a long (~5.5 km) and low angle terminus compared to its counterparts and a
significant portion of that tongue was located near sea level (Figure 12). This is due to the
subtle and flatter topography of eastern Heard Island (see also Figure 1c). Low angle
glaciers are particularly sensitive to climate change (e.g. Oerlemans, 2001) and those
that terminate at sea level are subject to rapid surface and subaqueous ablation
throughout the year due to warm air temperatures and melting by seawater and
mechanical calving by tidal and wave action (Truffer and Motyka, 2016). Therefore, we
propose that, along with increased air temperatures, the high retreat rate of the
Stephenson Glacier may be related to the low topography and flat surface of the glacier
tongue and the negative impact of the very large lagoon connected to ocean water. We
also note that despite recent efforts to quantify ocean water properties in similar
environments (Mortensen et al., 2013; Straneo and Cenedese, 2015), rates of
subaqueous melting for Heard Island glaciers along with the water properties of the
associated lagoons are largely unknown and require further investigation. Finally, the



small readvances observed at Stephenson's Glacier are also characteristic of tidewater
glaciers which can undergo non-climatic advance and retreat cycles (Vieli, 2011).

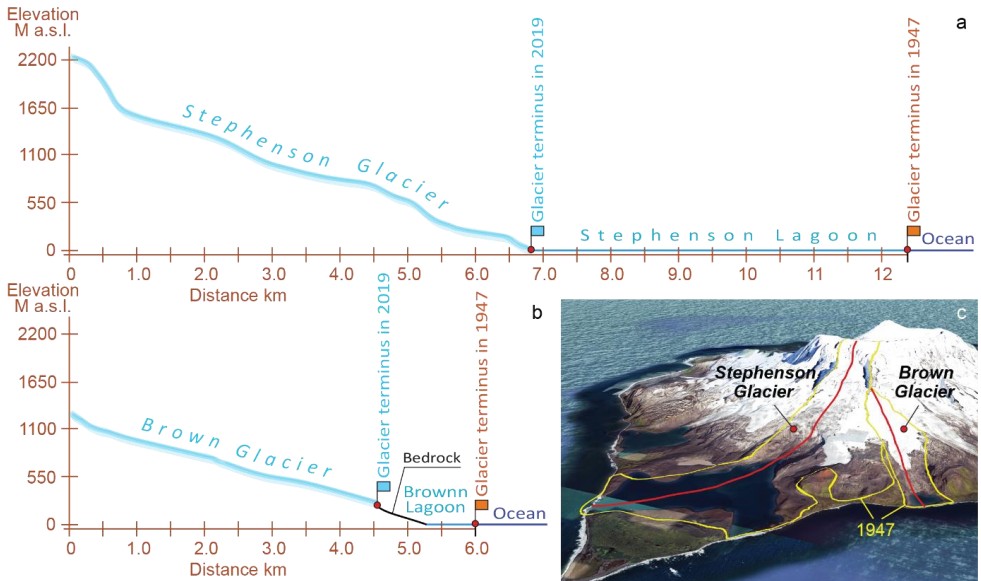

**Figure 12**. Longitudinal profiles based on Pléiades DSM, 2019. a – Stephenson Glacier; b
– Brown Glacier; c – Longitudinal profile paths (in red) for both glaciers on 3D view (©
Google Earth).

## 6.3 Impact of surface debris cover on glacier retreat

Despite an overall increase since 1988, surface debris has varied between individual
glaciers and some glaciers have contradicted the overall trend. e.g. A substantial amount
of surface debris cover was found on Stephenson Glacier in 1988 (~14%), mainly near the
terminus, which had almost disappeared by 2019 (~2.9%). This could be because the
glacier still had a low-angle tongue in 1988 that was favourable for debris accumulation
(e.g. Mölg et al., 2019), but that this debris-laden ice subsequently calved into the lake.
The debris cover at Stephenson Glacier prior to 1988 may have contributed to the
accelerated terminus retreat if debris was less than a few cm thick, as this would have
led to enhanced surface melting (e.g. Östrem, 1959). However, surface debris thickness
and properties have not been measured at Heard Island and this inference is speculative.
The most significant increase of surface debris cover was observed on northeast facing
Downes Glacier from ~9.8% in 1988 to ~31.6% in 2019, while the total area of this glacier
has only reduced by ~4.3% during the same period (Figure 3, 8). It is not clear what drove
this increase, though the lack of bare cliffs above the glacier (Figure 13) suggests a
subglacial source. More investigation is needed as there is no clear process that can



account for the observed development of surface debris cover at Downes Glacier relative
to other glaciers on Heard Island.

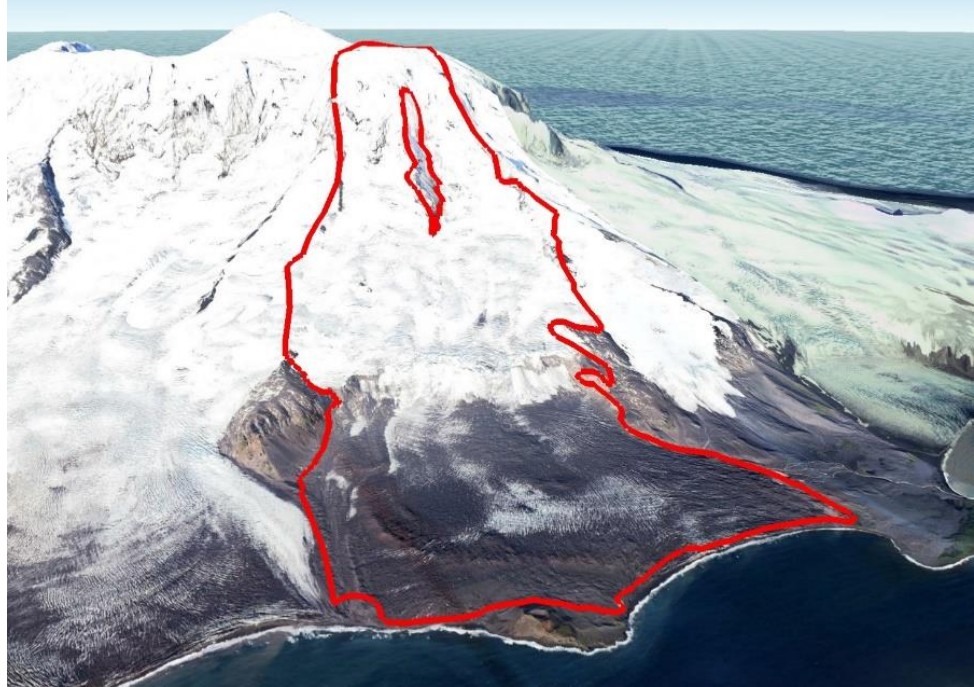


**Figure 13**. The most heavily debris-covered Downes Glacier within the Heard Island
(31/03/2024 © Google Earth).

A global assessment of surface debris cover by Scherler et al. (2018) uses RGIv6 outlines
for Heard Island delineated based on Spot images from 1988. We extracted these
outlines from Scherler et al. (2018) for our glacier sample from 1988 to compare results
(Figure 14). We found that Scherler et al. (2018) reported much higher percentage (25.2
%) of surface debris cover for Heard Island than our study (7.0) in 1988. This does not
even match within our uncertainty (±6.0 %). These differences are probably explained by
automatized method of global assessment of surface debris cover by Scherler et al.
(2018). Our approach uses accurate manual digitization, while automated mapping often
fails and manual corrections are required (e.g. Paul et al., 2013).



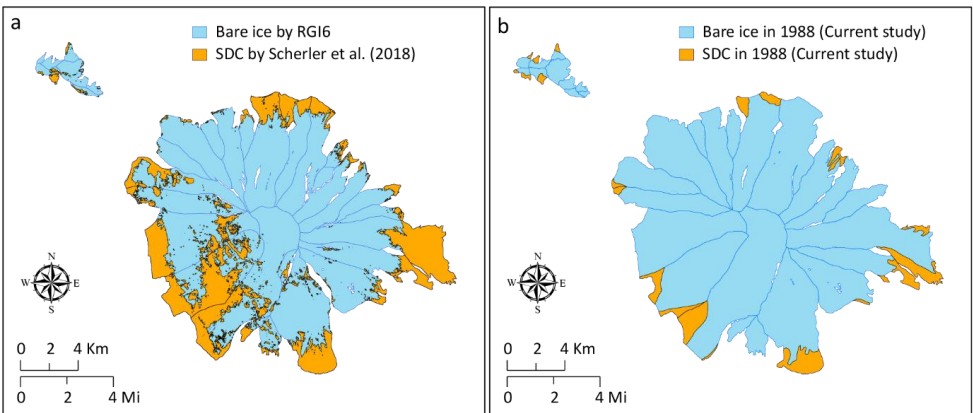

**Figure 14**. a – extracted surface debris cover (SDC) data from Scherler et al. (2018).
Glacier outlines from RGIv6 from 1988; b – SDC from our study from 1988.

## 7 Conclusions

We present a multi-year glacier inventory based on historic topographical maps,
medium-, and high-resolution satellite imagery and digital elevation/surface models. Our
findings show that glaciers on Heard Island experienced significant area loss (22% or
$-0.31\%$ yr$^{-1}$) over the 72-year period between 1947-2009, from 289.4±6.1 km$^2$ to
225.7±4.2 km$^2$ with an increasing rate of ice loss in recent decades. Our study also shows
that the small and low elevation glaciers at Laurens Peninsula have experienced the
highest recession rates of $-1.1\%$ yr$^{-1}$ over the last 72 years.

We observed much higher glacier area decrease and terminus retreat on the east and
southeast side of Heard Island. Although this asymmetry may have an underlying climate
driver, for example due to increased foehn winds, topographic factors have clearly played
a role; ice retreat on the eastern side of the island has been enhanced at low-elevation
and low-angle tongues where glaciers have transitioned from land based to lake calving.
Stephenson's Glacier is an exemplar of this behaviour. Surface debris cover has also
increased on most Heard Island glaciers and could also be a contributor to accelerating
ice loss.

Our new glacier inventory for Heard Island tracks changes in glacier extent, and
dynamics, providing insights into the impact of climate change in this remote and
sensitive region. Further work is required to test some of the ideas we have put forward in
this paper regarding the different roles of climatic and non-climatic factors in driving ice
retreat. The new inventory will also help to better understand how changes in glaciers
affect biodiversity and ecosystems, which will inform conservation and management
efforts at this World Heritage site.



**Data availability**

Will be submitted to GLIMS and can be used for future studies

**Author contributions**

LGT and ANM designed the conceptual framework for the study. LGT mapped glacier
outlines and wrote the paper with input and feedback from ANM and WY.

**Competing interests**

The contact author has declared that neither they nor their co-authors have any
competing interests.

**Acknowledgements**

Medium-resolution SPOT and high-resolution Pléiades images acquired by French Space
Agency (CNES)'s Spot World Heritage Programme and Pléiades Glacier Observatory. We
are very grateful to Dr Etienne Berthier for his help in obtaining the satellite images.

**Financial support**

This work was supported by the Australian Research Council (ARC) Special Research
Initiative (SRI) Securing Antarctica's Environmental Future (SR200100005).

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
