# Peer review of "Glacier inventories reveal an acceleration of Heard Island glacier loss over recent decades"

_EGUsphere, 2024_

## Author Comment (AC1)

**Authors reply to Dr. Frank Paul**

**"Glacier inventories reveal an acceleration of Heard Island glacier loss over recent decades"**

by L. G. Tielidze, et al.

EGUsphere

https://doi.org/10.5194/egusphere-2024-3811, in review, 2025

**Dear Dr. Frank Paul**,

First, we thank you for your careful reading of the paper and for the constructive review. In the following pages, we provide point-by-point responses following every comment.

All corrections and changes we made in the text are marked up using the "Track Changes" function.

Best regards,

Levan Tielidze on behalf of all co-authors

**General comments**

The study by Tielidze et al. presents glacier inventories from three points in time (1947, 1988, 2019) for Heard Island along with an analysis of glacier changes, partly also for denser time series. The study is also investigating debris cover evolution, glacier length changes, climatic fluctuations, mass balance time series and a special calving event of one glacier. In my view this is an important and well-written study that closes major knowledge gaps in this remote part of the world. Apart from the wrong interpretation of the calving event, I have many small comments to the text and figures. I think quite a lot can be and partly should be improved to get things right. Still, these are all small points that the authors can hopefully consider.

Thank you for the positive feedback and please see our comments regarding the incorrect interpretation of calving event and other relatively minor questions below. Please note that interpretation of the Stephenson Glacier retreat/readvance was also a topic of discussion among the co-authors prior to submission of the first manuscript in 2024. We are now satisfied that it is correctly represented.

My major objection is the interpretation of the 'advance' of Stephenson glacier in 2007/8 depicted in Figs. 7 and 11 and described in various parts of the text. I agree that it is

looking like an advance of the tongue, but actually the retreat is continuous and what is advancing / moving forward / expanding is just floating ice bergs from the disintegration of the tongue. As it is a bit difficult to see in Fig. 7, I have downloaded the panchromatic (15 m) bands of the Land- sat ETM+ scenes listed in Table 1 from 2001 to 2010 (plus two additional scenes from 2008) and then animated close-ups of the region at various speeds. I also flipped back and forth between two individual images to reveal the change.

Thanks for your help in interpreting this properly. We present the updated glacier retreat rates in Figure 11. We also edited text everywhere to incorporate the correct retreat rates.

[Figure]

**Figure 11**. Cumulative length changes between 1947–2019 for Brown and Stephenson glaciers (current study) compared to Agassiz and Chamonix glaciers on Kerguelen (Deline et al., 2024) and Tasman Glacier in New Zealand (Purdie et al., 2020; Mackintosh et al., 2017).

By 2007 most of the SE branch of the tongue disintegrated, leaving an arcshaped region at the glacier terminus. Between 2007 and 2008 there is very little change here but larger parts of eastern branch fell apart. The dense melange of icebergs is drifting to the sea, giving the impression that the glacier would advance. Flipping than back and forth between the 2008 images from 4.2. and 23.3. reveals that the upper part of the glacier tongue is static and does not show any movement, whereas the floating ice bergs continue their way outward. There is a little bit of tongue retreat from 2008 to 2009, but the major change is that all the floating icebergs melted away. This would not have been possible in just a year with a compact glacier tongue that had just advanced to this position. Please note that there is also no advance from 2001 to 2003. What looks like

an advance is also the movement of the floating icebergs from the disintegrating and further retreating tongue.

We agree with your comment here and reaffirm that we have changed the Stephenson Glacier retreat rates and related texts and figures throughout accordingly.

I suggest measuring the changes again and revising the text accordingly. One might add the full (annotated) 2001 to 2010 time series of Landsat pan images in the supplement and/or provide screenshots in a zip file so that interested readers can also follow the development.

Done. The text was revised accordingly everywhere, and supplementary materials (screenshots in a zip file) were added.

**Specific comments**

L40: There are further (less negative) values from some other, more recent studies. Maybe cite these as well?

Done.

L42: I suggest writing here 'glacier shrinkage' to clearly include mass loss; the retreat of a tongue has usually only very limited impact on water supplies.

Done.

L44/45: I suggest adding some further studies related to applications of glacier inventories, for example doi.org/10.1038/s41586-021-03436-z, doi.org/10.1038/s41558-017-0049-x or doi.org/10.1126/science.abo1324

Done.

L47: I suggest writing 'in combination with glacier outlines they provide crucial data …' The DEM alone does not proved any of these.

Done.

L65: I understand glacier retreat as a part of an ecosystem. An ecosystem should not be at risk from changing glacier extents. I suggest writing 'at risk from the impact of human settlements' as usually we are the problem.

We have explained further. Please also see the next response.

L66: I am a bit unsure how this could be important for a place like Heard Island? Do they have a problem with 'overtourism' or specifically protected areas? Please clarify. My understanding is that in this region nature (glaciers, volcanoes) prescribes the changes and human interference is - apart from some scientists - very limited.

Regarding both of these points - Glacier retreat causes previously unconnected ecological communities to come into contact, putting ecosystems at risk. And because Heard Island is largely free of human influence it is an ideal site to study such interactions. We have explained this process more completely and cited an appropriate paper on the global issue as well as the Heard Island action plan, which discuss these issues.

L74: I would suggest removing the 'why' question. Just from some sparse remote sensing data and some coarse resolution reanalysis data (but without any field measurements), I think our understanding of how the glaciers work in this region and why they are changing in the way they do is very limited. Adding geothermal heat flux, ash layers and ocean calving, I would rather say that we are quite far away from understanding anything here.

Done.

L97: of Heart Island (no 'the')

Done.

L136: What do other studies say about glacier flow velocities or ice thickness (e.g. doi.org/10.1038/s41561-021-00885-z)? Do they roughly agree?

We added the sentence about the ice velocity.

'A recent global study of glacier thickness and velocity (Millan et al., 2022) based on Landsat and Sentinel image pairs acquired between 2017 and 2018, shows unusually fast flow velocities for the Gotley Glacier with velocities up to 1,500 m a$^{-1}$.'

L145: What do today's global scale studies say about the glaciers of Heard Island, e.g. regarding geodetic mass balance and derived gradients or future evolution?

We discuss this in the '6.1 Glacier retreat and climate drivers'.

L163: Compared to glacier flow fields, the ice divides in RGI 6.0 do not look too bad in Fig. 1 of Millan et al. (2022). Please describe in the discussion section (or supplemental material) what the problems are.

We add the new section into the Discussion chapter with an appropriate illustration. Please see the Chapter 6.4

[Figure]

**Figure 14**. Comparison of RGI v6 (Pfeffer et al., 2014; Maussion et al., 2023) and our study for the Boudissin Glacier (a) and Downes/Ealey glaciers (b). Black arrows show inaccurate ice margins in RGI v6-v7 (due to misidentification of ice divides).

L188 (Fig. 2): I think panel d with the DEM is not required, at least not in this form. If at all, please show a hillshade version with elevation contours and glacier outline overlay. Apart from this, I have the impression that the images are a bit too small to see anything. If not in the main text, please show each image at full page width in the supplemental material.

We deleted panel 'd' and increased the resolution of the rest of the panels.

L194: Apart from cloud cover, I think also the commercial phase of Landsat was an issue. Images have just not been acquired in the 1990s.

Corrected accordingly.

'Landsat satellite products have been available since the 1980s, although image acquisition was limited in early 1990s due to its commercial phase (Williamson, 1997).

Post-1990s Landsat images of Heard Island often feature cloud cover that partially or completely obscures glaciers, making it difficult to accurately delineate glacier boundaries......'

L223: Aren't we at GDEM v3 in the meantime? And has the AW3D30 DEM also been

checked? At times it has a better quality than the GDEM. Please shortly comment on it. We double-checked and can confirm that our GDEM is v3. (Screenshot below). It is also corrected in the manuscript.

[Figure]

Regarding the AW3D30 DEM, we have access to PALSAR 30 m DEM, but we didn't use it as we also have much higher resolution Pleiades DEM from 2019.

To clarify again, we used ASTER DEM (2000) for the glaciers from 1988, and Pleiades DEM (2019) for the glaciers from 2019.

L239: I would not say that manual mapping is more suitable in general, but already add here that for a small study region with only a few glaciers that are often debris-covered, complete manual delineation is a similar fast approach.

We replaced 'suitable' with 'accurate'.

L245: When there is (optically thick) debris cover on the glacier surface is does not only create uncertainty, but this part is then simply not mapped.

Corrected accordingly.

L251: from days? This might be of interest for surging glaciers, but I think for all others annual resolution should be sufficient.

Replaced by 'years.

L257: Please repeat this exercise for Stephenson Glacier after the terminus region has been digitized correctly.

Done.

L269: When using the buffer method, please also mention that it only buffers the extents that are not ice-ice divides (where the method makes no sense). As many glaciers are debris-covered and manual delineation has been used in this study, please also apply the multiple independent digitizing method for uncertainty assessment, at least for the terminus section of about five glaciers. The results obtained for the uncertainty will be more reliable. And please apply the method to both, the Landsat and the Pleiades images.

Done. We applied multiple digitizing method for both, the SPOT and the Pleiades images. Please see the new text and the Figure.

'To estimate glacier mapping uncertainty, first we tested multiple digitization (e.g. Paul et al., 2013; Tielidze et al., 2022). A sub-sample of two glaciers from the medium-resolution SPOT image, 1988, were re-digitized by three different operators. The selected glaciers included debris-free Brown and one unnamed debris-covered, glacier with Glims ID of G073625E53121S. The uncertainty for the debris-free Brown Glacier based on normalized standard deviation (NSD – delineations by multiple digitalization divided by the mean glacier area for all outlines) was 2.0 %. In contrast, the debris-covered glacier exhibited a much higher uncertainty of 5.3%. We applied the same methodology to these glaciers using the high-resolution Pléiades image from 2019. The mapping uncertainty for the debris-free glacier was determined to be 1.1%, while the debris-covered glacier exhibited a considerably higher uncertainty of 5.1% (Figure 3a-d).'

[Figure]

**Figure 3**. An example of multiple digitization for debris-free Brown (a-b) and debris-covered G073625E53121S (b-c) glacier terminus. a-c - SPOT scene (09/01/1988). b-d - Pléiades scene (07/04/2019). Insert map on panel 'b' shows location of the selected glaciers relative to other glaciers.

L302 (Figs. 3b and c): I suggest removing these two plots as they only show what is already stated in the text and the trend is not too difficult to follow. Moreover, I dislike the fact that the 'Year' axis is not drawn to scale. The distances of the bars to the one from 1988 should be different at a 4:3 rate. Instead of these two panels, please show in a new Figure 4 a scatter plot with the area change rate per year (or decade) vs. size for each individual glacier, using different colours/symbols for the two periods and maybe also for the two islands. Is there a reason why the glacier south of Nr. 16 does not have a number?

 Done. Please see the new panels on page 12.

Panel 'a' was also modified and all glacier names/IDs have been added.

L305 (Fig. 4): Please remove the grey bars in the background, draw the years to scale and show boxplots representing the values for the individual glaciers in the background, maybe also colour coded (but please do not use red and green in the same plot red, blue and black might work instead).

We removed the grey bars and drew the years to scale. We did not add individual glacier values because it made the figure too complicated.

L313: Please explain why there is a downward shift of maximum elevation. This is not obvious from Fig. 3a where maximum elevation looks unchanged.

Explained in the text. This was because of the glaciers at Laurens Peninsula.

L323: I think for such a small number of glaciers the statistical distributions are more governed by chance than by climate (in particular the count). If the information should be kept, I suggest writing explicitly 'By number, most glaciers on Heard Island are oriented towards …, whereas most of the area is facing southwest' (or something similar).

This paragraph was edited accordingly.

'By number, most glaciers on Heard Island are oriented towards the northeast, whereas most of the area is facing southwest. Glaciers of west and southwest orientation have the highest mean elevations (Figure 6a-c).'

L337 (Fig. 5): Please indicate the zero point in the center with a small cross. If the graphs should be kept, please show them side-by-side. There is not too much info included so they can be shown smaller. Instead of showing absolute values in Figs. 5a and b, it would also be possible to present relative numbers, allowing a display of the two curves in the same plot.

Done. Figure was updated. Pleases see the new Figure.

[Figure]

**Figure 6**. Distribution of Heard Island glacier aspects by (a) number, (b) area (km$^2$) and (c) mean elevation (a.s.l.) in 2019.

L349 (Fig. 6): I think there is no need to show length changes here and in Fig. 11 and suggest removing them here. Please also consider that 'Terminus change' on the right y-axis should read 'Cumulative length change' and that the years on the x-axis should be drawn to scale with the 1947 value being far away from 1988 and 1988 being 13 years away from 2001. The current display could be seen as a suggestive data manipulation and is also inappropriate to reveal that the recent retreat rate of Brown Glacier is much higher than in the decades before. Finally, given that the changes of Stephenson Glacier will be much less exciting and because annually resolved area changes have limited glaciological meaning, I suggest skipping Fig. 6 altogether and show the length changes in Fig. 11 only.

Done. Fig 6 was removed. Fig 11 was updated accordingly. We provided the new Figure above.

L354-361: Please replace this section with what has really happened as described in the general comments (e.g. continuous disintegration of the tongue since 2001, movement of the dense pack of icebergs to the sea, seemingly indicating an advance, the rapid meltdown of all ice bergs in just one year).

Done. Please see the new paragraph:

'Stephenson Glacier retreated more than any other glacier on Heard Island during our study period. The glacier terminus retreated by 5811±20 m between 1947–2019 yielding an annual retreat rate of −80.7 m yr⁻¹ (Figure 7). The highest retreat of Stephenson Glacier with −5541±20 m was recorded during the more recent period (1988-2019), corresponding to an annual retreat rate of −178.7 m yr⁻¹ which is seven times higher than it was observed for Brown Glacier during the same time. Observations in Stephenson Lagoon during the summer of 2008 revealed a dense concentration of icebergs drifting towards the ocean, creating the illusion of a glacier readvance. However, the upper section of the glacier tongue showed no change. Although a small terminus retreat was recorded from 2008 to 2009, the primary change was the complete loss of the floating icebergs by 2009'

L364 (Fig. 7): Please replace with correct and thicker outlines, instead of the dark red use a brighter orange. Please add the full time series as supplemental material (maybe annotated images in a PDF and the images itself).

We deleted old figure and provided the new one that better shows glacier retreat according to the different years. The supplementary materials (screenshots in a zip file) were also added.

[Figure]

**Figure 7**. Stephenson (a) and Brown (b) glacier retreat between 1947 and 2019. Pléiades Hillshade is used as a background (07/04/2019). Insert map on panel 'b' shows location of the selected glaciers relative to other glaciers.

L367: I speculate here a bit, but as the source of the debris cover is likely by ash falls rather than by rock fall from ice-free headwalls, maybe this can be added in a short introduction to this section? Moreover, when trends are derived from just two images, the differences might only be related to different snow conditions rather than a real increase (e.g. when the snow line in 1988 was lower). Please mention this as well and consider it for the discussion.

Done. We added a paragraph about the tephra in the Discussion section (6.3).

'It is widely observed that in glacio-volcanic regions, volcanic eruptions regularly deposit tephra onto adjacent glaciers (Nield et al., 2013). These deposits into the ice system produces a significant component of the surface debris on these ice masses (Kirkbride and Dugmore, 2003). Ash-ice interactions can significantly modulate glacial mass balance responses to climatic changes, affecting glaciers for decades following eruptions (e.g. Richardson and Brook, 2010; Rivera et al., 2012). Given Heard Island's glacio-volcanic nature, we consider both rock fall and tephra deposition, or a combination, as possible debris cover sources.'

L382 (Fig. 8): Please show panels a and b side by side.

Done.

L393: with a previous record

L395: Why is the trend consistent and why 'glacier melting'? Are there any records of melting for the past 72 years? As far as I can see, only area and length changes are derived over this period and I would name this glacier retreat rather than melting, the latter being related to mass balance. As the relation between length and area changes is rather complex (e.g. depending on response times and ice thickness distribution), both length and area changes are difficult to link to climate. Although the values in Fig. 10 back to 1976 might be wrong, the low temperatures visible in Fig. 9 might indeed have caused positive mass balances until 1997 (except in a few years around 1981). Hence, I would expect stable or advancing rather than retreating glaciers from 1947 to 1988.

We correched this paragraph accordingly.

'Comparison between the first (1947–1959) and last (2009–2019) decades show summer temperature (NDJFM) increase by 0.7 °C which is consistent with Thost and Truffer (2008) who observed 0.8 °C increase of the summer temperatures between the 1948–1955 and 1997–2005 epochs. As shown in Figure 9, a warming trend of the mean annual summer temperatures and temperature anomalies relative to the long-term mean, specifically from the 1980s, also aligns with higher glacier retreat over the second investigated period, from the 1990s onwards.'

L402: I will not go into too much detail here, but I find this interpretation a bit rough. For example, albedo changes would only occur if the rainfall removes snow cover and exposes bare ice. However, with nearly permanent cloud cover there is little incoming shortwave radiation so that albedo changes should have a rather limited impact.
We have edited this slightly, removing reference to albedo. The effect of warming on the elevation of the snow/rain threshold and increasing turbulent fluxes is well established for maritime glaciers.

L403: Higher temperatures

Done.

L406: I can see the increase, but think it is more over the past 30 rather than 70 years. Excluding the years around the 1980s, temperatures seem to be constantly low from

1947 to1995 (or at least until 1980) and really rise only afterwards. As mentioned above, I would thus not expect much or constant retreat from 1947 to 1988.

Corrected as 'over the past 30'.

L414: Is this speculation about mass balance required? Given that the climate-glacier relation is not well known for this region and several glaciers terminate in lakes (one revealing a calving instability) or the ocean, I would not derive from just two observations of glacier extent what the governing mass balance might have been, in particular not for the same period, as response times have to be considered.

Corrected. We use 'area loss' instead of 'mass balance'. We are also more circumspect in the discussion with respect to glacier length changes and climate drivers, and mention several possibilities, including a lagged response or data issues.

L416: How do the corresponding elevation change maps look like? Are there any data gaps? I would also not plot linearized annual values when only 5-year means make sense. This could also be seen as a data manipulation (also because it is not described how the original data have been converted). Please show 5-year mean values (maybe with an uncertainty range) as a straight line covering the respective periods instead.

We modified this paragraph. We also update the Figure.

[Figure]

**Figure 10**. Mean annual geodetic mass balance (m w.e. a$^{-1}$) and cumulative mass balance (m w.e.) for all Heard Island glaciers between 2000-2019 (Hugonnet et al., 2021) (averaged within a rolling window of 5 years) and between 1976–2019 (Dussaillant et al., 2025).

L418/419: This might be correct, but as mentioned above, I would not directly link observed area trends to the governing mass balance forcing.

We corrected this paragraph:

'Our new glacier inventories indicate that Heard Island glaciers experienced higher area loss between 1988 and 2019. This finding is supported by satellite-based geodetic mass balance estimates for all Heard Island glaciers extracted from the global study of Hugonnet et al. (2021) (Figure 10). Negative mass balance values between 1990s and 2019, derived from a mixed method that incorporates in situ observations (Dussaillant et al., 2025) also aligns with higher glacier area loss from the 1990s onwards. A positive geodetic mass balance trend between 1976 and 1990s (Dussaillant et al., 2025), perhaps due to a serious of relatively cooler years (Figure 9), contrasts with our observed glacier area reduction during the same period. This mismatch (Figure 10) could be for several reasons: it might be because the glacier response is lagged and is responding to earlier warming. Or it could be simply a function of data limitations including a lack of direct mass balance observations to constrain the estimates of Dussaillant et al. (2025) and limited instrumental meteorological observation in this remote region to support the ERA5 dataset.'

L429: increasing air temperatures

Done.

L426: Why should the shift be human-induced? Do oceanic oscillations care for human activities?

We deleted 'human-induced' to avoid confusion. We also added the new sentence:

'This shift has been attributed to the combined anthropogenic effects of increasing greenhouse gases and decreasing stratospheric ozone (e.g. Son et al., 2008).'

L429: I am unsure if this conclusion can be made here. The glaciers investigated by Bakke et al. on South Georgia are rather different (flat cirques, calving instability) than those on Heard Island (mostly originating at a high volcano). I am also unsure what the issue should be with the impact of retreating glaciers on the landscape etc. The glaciers do also impact the landscape when they are present. Also the irreversibility is unclear to me. Why should glaciers not come back when it is getting colder again?

We removed this sentence.

L434 & 442: Again, I think you mean here glacier mass loss rather than retreat.

Yes, corrected.

L442: Due to the insolating effect, the deposition of tephra might decrease glacier mass loss.

Yes, we added an appropriate sentence here. We also deleted some speculations and extend the discussion – 6.3.

'Volcanic activity is another potential factor that could be contributing to the accelerated glacier melting on Heard Island. Glacier mass loss also be a trigger for enhanced volcanism at Heard Island due to decompression of the underlying magma chamber (e.g. Barr et al., 2018). Eruptions could cause significant melting of the ice or deposition of the thin layer of tephra, while increases in geothermal heat may lead to greater basal melting, thereby influencing glacier movement (Allison and Keage, 1986; Fox et al., 2021). Conversely, sufficiently thick deposition of tephra on glacier surface might also decrease glacier mass loss (Kirkbride and Dugmore, 2003). However, given the lack of direct evidence in satellite images for unusual melt or tephra cover, the fact that accelerated glacier melt occurred on all Heard Island glaciers and not just a subset, and also on nearby Kerguelen Islands (Berthier et al., 2009; Deline et al., 2024), we consider that any volcanic drivers of glacier retreat have been less important than climatic ones during our observation period.'

L446 (Fig. 9): Please remove panel a and add the scale with the real values to the right of the plot in panel b. There is no need to show the same curve twice. I would also repeat major tick marks on the lower x-axis. Currently the values are quite far away from the marks. One might also add major grid lines for the x-axis and minor tick marks for each year.

Done.

L449: I think there should be a space between the 0.7 and °C as well as between 2 and m.

Done.

L454 (Fig. 10): As mentioned above, I would replace the linear mass balance trend from Hugonnet et al. with the real 5-year averages for the region.

Done.

L460: sparse

Done.

L462: Isn't the volcano cone a bit too narrow (in its higher reaches) to create foehn winds, i.e. force the air to rise, or reduce precipitation? On the satellite images the air seems quite often to flow around the cone rather than over it. For the Cook Ice Cap on Kerguelen Island (which is much wider) the situation is rather different. And as described in L471ff the real driver for rapid retreat of glaciers 1, 14, 15 and 16 seems to be lake formation and calving.

We edited these sentences and use 'orographic effect' instead of 'foehn wind'. The orographic effects on Heard Island climate are well established and are reflected in the limited station data available from Atlas Cove and from near Stephenson's and Brown glaciers.

L476: Apart from the fact that a linear retreat rate cannot be shown in Fig. 11 as it has a highly non-linear scale, where is the linear retreat rate of Chamonix Glacier? From 2008 to 2019 it is a more or less straight line.

We updated this Figure and added the linear scale. The new Figure 11 is provided above.

L477: I think one should not directly compare retreat rates of Fox and Franz-Josef Glaciers at the selective temporal and extended length change scale presented here. Both had strong advances after 1980, which are supressed here as values from 1988 to 2001 are not shown. Please use a linear scale for the years.

Corrected. Fox and Franz-Josef glaciers are removed now. Linear scale was corrected.

L489 (Fig. 11): As mentioned above, please use a linear scale for the year (or only show a subset, e.g. start in 1988). Please also place the x-axis at the bottom (as in all other plots).

Done. Please see our response and the new Figure above.

L498: I think Stephenson is in its accumulation region very steep and thus not a low angle glacier. A small change of the ELA would only cause a small change in the size of the accumulation region, i.e. result in rather insensitive glaciers. What matters here is that the accumulation region is narrowing towards the top which is basically the opposite for usual valley glaciers such as Fox or Franz-Josef. A flat lower part can be sustained for quite some time as long as it is land terminating. The important point for the rapid retreat of Stephenson Glacier should thus be the lake formation and calving instability.

We edited this section accordingly. Please see:

'The retreat of Stephenson Glacier warrants further discussion. In the 1950s, the glacier had a steep and small accumulation area relative to its long (~5.5 km) and low angle terminus compared to its counterparts. A significant portion of that tongue was located near sea level (Figure 12). This is due to the subtle and flatter topography of eastern lower part of Heard Island (see also Figure 1c). This steep/small accumulation area, low angle terminus, relatively high ablation rates at sea level and melting by seawater and mechanical calving by tidal action (Truffer and Motyka, 2016) likely made the glacier more susceptible to warming. Once lagoon formation initiated, the high retreat rate of the Stephenson Glacier was enhanced by calving as the glacier retreated into its overdeepend basin. We also note that despite recent efforts to quantify water properties in similar environments (Mortensen et al., 2013; Straneo and Cenedese, 2015), rates of subaqueous melting for Heard Island glaciers along with the water properties of the associated lagoons are largely unknown and require further investigation.'

L501: For glaciers 4, 5, and 6 'wave action' is obviously not a problem.

We deleted 'wave action'.

L503/4: In effect, it is a calving instability caused by a likely overdeepend glacier bed filled with (relatively warm) water.

We edited this section to refer more specifically to these processes. Please see the paragraph above.

L509: There was actually neither a small nor a large re-advance, this impression just resulted from flowing ice bergs (see general comments).

This sentence was deleted.

L513 (Fig. 12): I am a bit unsure if this figure is adding anything to the science. I suggest removing it here and moving it to the supplemental material.

We would leave it in the main text as it shows both glacier profiles.

L517: This section 6.3 is not really about the impact of debris cover on glacier retreat, but presents two contrasting trends of decrease/increase in debris cover. I think assessing the impact is not really possible from the data shown. Please rename.

Done. We changed the title as: '6.3 Evolution of Surface Debris Cover in 1988-2019'

L518-527: I think this is a bit trivial. Indeed, when a glacier is loosing its debris-covered tongue into a lake, the amount of debris cover will decrease. As indicated in L527, I would avoid speculating about debris-cover properties (it could also be a thin insulating ash layer) and please stick to the data derived in this study.

We removed this speculation about the role of debris cover in glacier change, and just report the values.

L519: e.g. a substantial

Done.

L531: The long bare ice cliff in Fig. 13 is in contrast to what is written here. Please check if the increase in debris cover is a result of a higher snowline in 2019 compared to 1988. It is difficult to see from Figs. 2b/c, but it looks like this.

We modified this paragraph accordingly:

'The most significant increase of surface debris cover was observed on northeast facing Downes Glacier from ~9.8% in 1988 to ~31.6% in 2019, while the total area of this glacier has only reduced by ~4.3% during the same period (Figure 4, 8). It is not clear what drove this increase (Figure 13). However, as our debris cover trends are derived from only two satellite images, the differences in debris-covered area might only be related to different snow conditions rather than a real increase (e.g. if the snow line in 1988 was lower). More investigation is needed as there is no obvious process that can account for the observed development of surface debris cover at Downes Glacier relative to other glaciers on Heard Island.'

L537 (Fig. 13): I wonder why a Google Earth perspective view is shown (from which date is the image?) when a DEM and very high-resolution image from Pléiades is available to the authors? Maybe show perspective view images from 1988 and 2019 side-by-side, annotated with the outline of debris cover and the position of the snow line? This would also bring the observations closer to the text. Currently Fig. 14b is only showing the 1988 status, where is the change to 2019?

The old figure was deleted, and the new one with two panels (1988-2019) was created based on your suggestion. Debris cover is marked with different colour. Please see the new Figure:

[Figure]

**Figure 13**. The most heavily debris-covered Downes Glacier along with Ealey Glacier within the Heard Island. a – 1988; b – 2019. SPOT_1 (09/01/1988) and Pléiades ortho image (07/04/2019) is used as a background. Insert map into the legend shows location of the selected glaciers relative to other glaciers.

L547: I think Scherler et al. have not mapped debris cover but clean ice (which is straight forward). They then subtracted the clean ice map from the RGI extents which include manually mapped debris-cover. Their larger debris extent might be due to cloud cover, maybe shadows and old extents (with a much larger Stephenson glacier) used in RGI 6.0.

We edited this paragraph accordingly.

'A global assessment of surface debris cover by Scherler et al. (2018) uses residual between RGIv6 outlines (which include manually mapped debris-cover) and an

automatically digitised clean ice map. We extracted these outlines from Scherler et al. (2018) for our glacier sample from 1988 to compare results (Figure 15). We found that Scherler et al. (2018) reported much higher percentage (25.2 %) of surface debris cover for Heard Island than our study (7.0 %) in 1988. This does not even fit within our uncertainty (±6.0 %). These differences are probably explained by automatized method of global assessment of clean ice by Scherler et al. (2018). E.g. Common problems in ice-cover mapping from optical imagery are related to shadows as well as cloud and snow cover and manual corrections are required (e.g. Paul et al., 2013), while our approach only uses accurate manual digitization.'

L556: I would call the sensor used high and very high resolution, medium is MODIS.

Corrected accordingly.

L564-569: Please check if these statements are still valid and revise them accordingly

Done.

L 571: What is meant here with dynamics that is not 'changes in extent'?

Corrected accordingly.

'Our new glacier inventory for Heard Island tracks changes in glacier extent providing insights into the impact of climate change in this remote and sensitive region. Further work is required to test some of the ideas we have put forward in this paper regarding the different roles of climatic and non-climatic factors in driving ice retreat. The new inventory will also help to better understand how changes in glaciers affect biodiversity and ecosystems, which will inform conservation and management efforts at this World Heritage site.'

References

Please apply a consistent style to the references, e.g. for citing doi's. Now it is a wild mix.

Done.

---

## Author Comment (AC2)

**Authors reply to Anonymous Referee**

**"Glacier inventories reveal an acceleration of Heard Island glacier loss over recent decades"**

by L. G. Tielidze, et al.

EGUsphere

https://doi.org/10.5194/egusphere-2024-3811, in review, 2025

**Dear Referee**,

Thank you very much for your comments which we help to increase the quality of our manuscript. Please find in the following a point-by-point reply to your review.

All corrections and changes we made in the text are marked up using the "Track Changes" function.

Best regards,

Levan Tielidze on behalf of all co-authors

The study by Tielidze et al. presents a comprehensive analysis of glacier changes on Heard Island over multiple decades, utilizing historical topographical maps and satellite imagery. The study effectively highlights trends in glacier retreat, debris cover evolution, and climate influences. However, certain aspects require further clarification and refinement to strengthen the conclusions drawn.

Thank you for positive feedback and please see our detailed comments below.

My major concern is about Stephenson Glacier. Figure 7: The digitised boundaries showing retreat and advancement using three lines for 2005-08-09 needs to be confirmed with more data. For me, the area depicted by 2009 and 2008 is resultant of calving of icebergs. Can you recheck this with more data? The authors mention a rapid lagoon formation, but how does this compare with similar cases elsewhere? Could lake expansion and calving instability be primary retreat drivers rather than just air temperature increases? Stephenson Glacier has lost debris cover over time, which contradicts the general trend. Why is this the case? Does this suggest significant ice thinning or enhanced calving?

We remapped Stephenson glacier terminus and reinterpreted its changes. We also updated figure 7 and 11.

[Figure]

**Figure 7**. Stephenson (a) and Brown (b) glacier retreat between 1947 and 2019. Pléiades Hillshade is used as a background (07/04/2019). Insert map on panel 'b' shows location of the selected glaciers relative to other glaciers.

[Figure]

**Figure 11.** Cumulative length changes between 1947–2019 for Brown and Stephenson glaciers (current study) compared to Agassiz and Chamonix glaciers on Kerguelen (Deline et al., 2024) and Tasman Glacier in New Zealand (Purdie et al., 2020; Mackintosh et al., 2017).

'The retreat of Stephenson Glacier warrants further discussion. In the 1950s, the glacier had a steep and small accumulation area relative to its long (~5.5 km) and low angle

terminus compared to its counterparts. A significant portion of that tongue was located near sea level (Figure 12). This is due to the subtle and flatter topography of eastern lower part of Heard Island (see also Figure 1c). This steep/small accumulation area, low angle terminus, relatively high ablation rates at sea level and melting by seawater and mechanical calving by tidal action (Truffer and Motyka, 2016) likely made the glacier more susceptible to warming. Once lagoon formation initiated, the high retreat rate of the Stephenson Glacier was enhanced by calving as the glacier retreated into its overdeepend basin. We also note that despite recent efforts to quantify water properties in similar environments (Mortensen et al., 2013; Straneo and Cenedese, 2015), rates of subaqueous melting for Heard Island glaciers along with the water properties of the associated lagoons are largely unknown and require further investigation.'

Line 25-27: The increase in debris cover from 7% to 12.8% is mentioned, but its implications for glacier mass balance should be briefly noted.

It would have been a speculative because our knowledge for debris cover and its implications for glacier mass balance is limited. We only present more evidence-based results.

Line 30-31: The phrase "many questions about the behaviour of Heard Island glaciers remain unanswered" is too vague. Specify which key uncertainties remain.

This sentence was deleted.

Line 239: Manual mapping is stated to be "more suitable" than automated methods. However, this should be qualified—manual delineation is more appropriate for small regions with significant debris cover, but automated methods may be effective for clean ice surfaces.

We edited this paragraph:

'Despite some advantages of the automated mapping method of clean ice (e.g. Paul et al., 2013), manual mapping of glaciers is more accurate for many mountain regions around the world (e.g. Stokes et al., 2013; Nagai et al., 2016; Tielidze and Wheate, 2018; Korneva et al., 2024). In this study, glacier boundaries were, therefore, delineated manually. This was also mainly due to the i) unavailability of cloud-free satellite channels/bands for different years for the entire study area, which limited us to use different band ratio segmentation methods for automated mapping; ii) significant amount of debris-cover, which can cause uncertainty during automatic mapping or

hinder mapping; and iii) a relatively small study area, which was less time expensive than it would have been for an entire mountain range.'

Line 269: The uncertainty assessment should clarify whether multiple independent digitizations were conducted for validation.

We have carried out multiple digitizations. A new paragraph and figure have also been added:

'To estimate glacier mapping uncertainty, first we tested multiple digitization (e.g. Paul et al., 2013; Tielidze et al., 2022). A sub-sample of two glaciers from the medium-resolution SPOT image, 1988, were re-digitized by three different operators. The selected glaciers included debris-free Brown and one unnamed debris-covered, glacier with Glims ID of G073625E53121S. The uncertainty for the debris-free Brown Glacier based on normalized standard deviation (NSD – delineations by multiple digitalization divided by the mean glacier area for all outlines) was 2.0 %. In contrast, the debris-covered glacier exhibited a much higher uncertainty of 5.3%. We applied the same methodology to these glaciers using the high-resolution Pléiades image from 2019. The mapping uncertainty for the debris-free glacier was determined to be 1.1%, while the debris-covered glacier exhibited a considerably higher uncertainty of 5.1% (Figure 3a-d).'

[Figure]

**Figure 3**. An example of multiple digitization for debris-free Brown (a-b) and debris-covered G073625E53121S (b-c) glacier terminus. a-c - SPOT scene (09/01/1988). b-d - Pléiades scene (07/04/2019). Insert map on panel 'b' shows location of the selected glaciers relative to other glaciers.

Line 251: Terminus measurements are described, but the frequency of available images (e.g., annual vs. decadal) should be stated explicitly.

We edited this paragraph:

'Accurately quantifying changes in glacier termini is essential for effective monitoring of glacier changes over various timescales, from years to centuries. Methods for this technique each offer advantages and limitations (Lea et al., 2014). In this study, we only measured two glacier (Stephenson and Brown) lengths based on the Global Land Ice Measurements from Space (GLIMS) guidelines (www.glims.com) and by following Purdie et al. (2014). The flow direction of the glacier was manually determined to be perpendicular to altitude contours. We assessed terminus changes by comparing glacier outlines from different dates along the ice front, oriented perpendicular to the flow. Elevation changes at the glacier fronts were also measured at the intersection points.'

Please also see Table 1 for all available images that were used for terminus measurements.

Fig. 3: The authors highlight higher retreat rates on the eastern side of Heard Island. Would it be possible to overlay temperature or precipitation data to visually reinforce this finding?

We updated Figure 3. Please see the new Figure 4.

We also note that instrumental measurements of temperature and precipitation are rare from Heard Island. Therefore, we use the ERA5 dataset, which is also not high enough resolution to overlay the western and eastern sides separately. Instead, we expanded the discussion to explain this process more clearly.

'We observed that higher rates of glacier retreat occurred on the eastern side of the island (Figure 4). Although climate data are sparse and a process-based modelling approach is required for further investigation, we expect this greater sensitivity of east-facing glaciers is due to orographic effects; satellite observations and limited climate station data show that the eastern side of the island, in the lee of the prevailing westerly winds, is less cloudy, warmer and receives less precipitation. This asymmetry has been noted in previous studies of Heard Island glaciers (e.g. Thost and Truffer, 2008) and is consistent with other islands in the sub-Antarctic region where strong westerly winds

interact with mountain barriers. For example, Berthier et al. (2009) observed that the eastern part of Cook Ice Cap on Kerguelen shrank 2.5 times more (~28%) than the western part (~11%) between 1963 and 2003. Similar observations have been made in South Georgia, where glacier decrease on the east coast was higher than on the windy and wet southwest coast during the second half of the 20th century (Gordon et al., 2008).'

Line 403: The impact of decreasing surface albedo is mentioned, but cloud cover over Heard Island is persistent. Would reduced albedo have the same effect under high cloud conditions?

We deleted albedo and edited this paragraph:

'Maritime glaciers such as those on Heard Island are particularly sensitive to air temperature changes (e.g. Anderson and Mackintosh, 2006; Davies et al., 2014). Warming causes the frequent precipitation to fall as rain at higher elevation rather than as snow. Higher temperatures also increase melt rates due to strong turbulent exchange in these windy environments (e.g. Anderson et al., 2010; Anderson and Mackintosh, 2012). For these reasons, the observed summer temperature increases since 1980s are likely an important driver of ongoing glacier recession on Heard Island. However, few direct precipitation and mass balance measurements exist for Heard Island (Thost and Truffer, 2008), and Favier et al. (2016) showed that retreat of the Cook Ice Cap on the relatively nearby Kerguelen Islands is likely due to atmospheric drying since the 1960s rather than atmospheric warming. Further observations and modelling are required to increase confidence in our interpretation that climate warming was largely responsible for driving recent glacier retreat on Heard Island.'

Line 426: The discussion of human-induced warming should be carefully phrased. While anthropogenic climate change is a major factor, direct human interference on Heard Island is minimal.

We edited this paragraph accordingly:

'The warming trend evident in ERA5 data in the Heard Island region (0.7°C) is relatively small to the observed 1.1°C warming for the neighbouring Kerguelen Island between 1951-2020 (Nel et al., 2023) but it is in agreement with general Southern Ocean warming trend since 1950s (Auger et al., 2021; Li et al., 2023). Atmospheric and ocean warming in this region is associated with a shift towards the positive phase of the Southern Annular Mode (Pohl et al., 2021), including an intensification and poleward shift of the Southern Hemisphere westerly winds (Perren et al., 2020). This shift has been attributed to the combined anthropogenic effects of increasing greenhouse gases and decreasing stratospheric ozone (e.g. Son et al., 2008).'